# Maternal RNA transcription in *Dlk1-Dio3* domain is critical for proper development of the mouse placental vasculature
Ximeijia Zhang, Hongjuan He, Haoran Yu, Xiangqi Teng, Ziwen Wang, Chenghao Li, Jiahang Li, Haopeng Yang, Jiwei Shen, Tong Wu, Fengwei Zhang, Yan Zhang & Qiong Wu ✉

The placenta is a unique organ for ensuring normal embryonic growth in the uterine. Here, we found that maternal RNA transcription in *Dlk1-Dio3* imprinted domain is essential for placentation. PolyA signals were inserted into *Gtl2* to establish a mouse model to prevent the expression of maternal RNAs in the domain. The maternal allele knock-in (MKI) and homozygous (HOMO) placentas showed an expanded junctional zone, reduced labyrinth and poor vasculature impacting both fetal and maternal blood spaces. The MKI and HOMO models displayed dysregulated gene expression in the *Dlk1-Dio3* domain. In situ hybridization detected *Dlk1, Gtl2, Rtl1*, *miR-127* and *Rian* dysregulated in the labyrinth vasculature. MKI and HOMO induced *Dlk1* to lose imprinting, and DNA methylation changes of IG-DMR and *Gtl2*-DMR, leading to abnormal gene expression, while the above changes didn't occur in paternal allele knock-in placentas. These findings demonstrate that maternal RNAs in the *Dlk1-Dio3* domain are involved in placental vasculature, regulating gene expression, imprinting status and DNA methylation.

The mammal placenta locates between the maternal tissue and fetus, providing the necessary support and instructing the development of fetus in the uterine environment. Proper development of the placenta is essential to its functions, including nutrition exchange, providing an immunological barrier for the fetus, and so on. The circulatory system between mother and fetus exchanges substance in placental labyrinth area whose formation consists of a series of morphogenetic events, mainly the one that the allantois derived from extra-embryonic mesodermal cells contacts chorion around embryonic day 8.5 (E8.5). Then allantois-derived cells form endothelial cells and undergo extensive intertwining and branching with chorionic trophoblast cells to develop a highly branched vascular network of the mature placenta at about E12.5[1]. Any defects in any type of cells making up the placenta correlate strongly with embryonic viability, development and growth, especially at mid-gestation[2]. A study on large-scale gene knockout mouse lines has demonstrated that mutant mice with intrauterine lethality always exhibit placental dysmorphologies[3].

Genomic imprinting is inextricably linked to mammalian placentation[4]. Imprinting genes that are monoallelically expressed in a parent-of-origin-specific manner can produce signals to modulate fetal developmental progress, as well as maternal resource allocation during pregnancy[5,6]. The *Dlk1-Dio3* imprinted domain spans about 1 Mb on mouse chromosome 12 (chr12) and human chr14, and contains three paternally expressed protein-coding genes (PEGs, *Dlk1*, *Rtl1*, and *Dio3*), multiple maternally expressed noncoding RNAs (MEGs, *Gtl2/Meg3*, *Rian/Meg8*, *Mirg*), and the largest known placental mammal-specific microRNA cluster[7]. Germline intergenic differentially methylated region (IG-DMR)[8] functions as the main imprinting control region (ICR) of *Dlk1-Dio3* imprinted domain and somatic *Gtl2*-DMR[9], *Dlk1*-DMR[10], *Meg8*-DMR[11–13] control gene expression together. *Dlk1-Dio3* imprinted domain plays crucial roles in many diseases, postnatal survival, as well as embryonic and placental development[14,15]. Numerous functions of *Dlk1-Dio3* domain about development have been demonstrated by developmental defects observed in individual locus component gain- or loss-model mice[16–23]. The imprinted genes in the *Dlk1-Dio3* domain are widely expressed in mouse placenta[19,24,25], the importance of *Rtl1* and *miRNA-127* in maintaining the fetal capillary network of feto-maternal interface have been revealed clearly[20,21], and others' functions remain elusive, especially MEGs.

Gene ablation study is a standard method for evaluating protein-coding or noncoding RNA gene function in vivo. Interestingly, different even opposite phenotypes in different model mice are observed when different gene knockout approaches are used to delete maternally expressed lncRNA *Gtl2* promoter and several different exons[9,26,27]. With different gene

School of Life Science and Technology, State Key Laboratory of Urban Water Resource and Environment, Harbin Institute of Technology, Harbin 150006 Heilongjiang, China. ✉e-mail: kigo@hit.edu.cn

editing methods adopted, gene expression in *Dlk1-Dio3* domain are quite different in those studies and the gene expression changes of those knockout mice cannot eliminate the destruction of epigenetic regulatory elements (like *Gtl2*-DMR) or chromatin spatial conformation. On the other hand, all those *Gtl2* knock-out mice studies didn't pay much attention to the placenta. Although deletion of *Gtl2* resulted in embryonic lethality after E12.5, it is still unknown whether *Gtl2* knock-out can affect the development of placenta, and then affect development and survival of the embryo during pregnancy.

Most MEGs at the *Gtl2-Rian-Mirg* locus in the *Dlk1-Dio3* domain are transcribed from a large polycistronic transcription unit and then regulated by the promoter of *Gtl2*[28]. The dysregulation of miRNAs from the locus leads to neonatal lethality[23,29], but these miRNAs and lncRNAs were overlooked in extra-embryonic tissue in past studies.

To explore the biological roles of those MEGs in placental development, we established a mouse model without eliminating any genomic sequence nor destroying any DNA regulatory element to reduce the transcripts of *Gtl2* and other MEGs in the *Dlk1-Dio3* domain. Through the model mice, we found that maternal RNA transcription functions in vasculature formation within the placental labyrinth zone and that ablation of MEGs in the placenta dysregulates the *Dlk1-Dio3* domain in gene expression, imprinting status and DNA methylation. Our research provides a new resource for understanding the functions and regulation of *Dlk1-Dio3* domain in the development of placenta. The conclusions will help future studies of mouse and human prenatal pathologies.

## Results

### Establishment of *Gtl2* polyA knock-in mouse model and analysis of placental weights

To better evaluate the regulation mechanism and functions of *Gtl2* and other MEGs in *Dlk1-Dio3* imprinted domain, we chose Easi-CRISPR genome editing technology[30] to insert three polyadenylation cassettes (3x polyA, 147 bp) into the end of exon1 of *Gtl2* to generate *Gtl2* polyA knock-in mouse model, without eliminating or replacing any genomic segments to prevent maternal RNA to transcribe (Fig. 1a). Almost all maternal RNA transcriptions in the *Dlk1-Dio3* domain transcribe from a large polycistronic transcription[28]. The promoter of *Gtl2* can initiate the expression of almost all maternal RNA transcriptions. The insertion site selected was located at the end of *Gtl2* promoter region, avoiding almost all epigenetic regulatory elements and miRNAs. Insertion of transcription termination signal (3x polyA) could cause early termination of the nascent lncRNA transcripts[31].

There were three ways to get four genotypic placentas for the research: Female heterozygous with *Gtl2* polyA knock-in (HET) were mated with wild-type (WT) males to get maternally *Gtl2* polyA knock-in (MKI) and WT placentas; WT females were mated with HET males to get paternally *Gtl2* polyA knock-in (PKI) and WT placentas; HET females were mated with HET males to get homozygous (HOMO) and WT ones. Genomic PCR was used to identify mutant mice (Fig. 1b). MKI and HOMO mice exhibited embryonic lethality in mid-gestation (around E13.5 to E15.5), while PKI pups could be born as normal ones (Supplementary Table 1). There was no significant difference in the gross appearance of yolk sac of alive embryos at different developmental stages. While, at E15.5, almost all MKI and HOMO embryos had died, and the blood vessels on the yolk sac were no longer seen clearly (Supplementary Fig. 1). For placentas, on the labyrinth sides, some blood vessels were thicker and more obvious in MKI and HOMO placentas than in WT ones. The PKI placentas were similar to WT ones during pregnancy. When all the MKI and HOMO embryos had been dead at E16.5, their labyrinth faces were pale and not full of blood (Fig. 1c). This phenomenon may result from the death of the embryo, and the embryo needs no longer a placenta to provide functions and the placenta degenerates slowly.

Then, we examined the placental weights of the surviving *Gtl2* polyA knock-in mice at E12.5, E14.5, and E16.5 (Fig. 1d–f). At E12.5 when almost all the embryos were alive, the MKI placentas almost had no changes compared to WT placentas, while the HOMO mutants had a little heavier in mean weight without statistically significant difference. As the development progressed to E14.5 when half MKI and HOMO embryos were dead, there were trends of weight increase in both the MKI and HOMO placentas, only HOMO ones had a significant change, compared to the control. PKI ones had similar weights to WT ones at any stage. MKI and HOMO placentas had no remarkable differences during the period before death, which suggests that MKI and HOMO placenta may be not responsible for embryonic lethality. The elevating changes in weight may affect the development of the placenta itself.

### *Gtl2* polyA knock-in alters placental architecture and leads to vasculature defect in the labyrinth

To further investigate histological change of *Gtl2* polyA knock-in placentas, placental morphology was observed at different stages. The mature mouse placenta is mainly divided into three layers, maternal decidua, junctional zone, and fetal labyrinth. The three-layer structure can be easily distinguished from each other according to the cell types of each layer[32]. To do the detailed statistical analysis of the proportions of three-layer areas, we visualized the expression of the junctional zone marker gene, *Tpbpa*[33], which expresses in the spongiotrophoblasts and glycogen trophoblasts and is used widely to recognize the intermediate junctional zone[34]. Then, we quantified each layer area, as development progressed, the proportion of junctional zone to the placental area gradually decreased and the proportion of labyrinth to the placental area gradually increased in the WT placentas. Compared to the control, MKI and HOMO placentas had bigger proportions of junctional zone and smaller proportions of labyrinth at E14.5 (Fig. 2a–c). Those changes in placental architecture began to occur at E14.5 and did not exist at E12.5, while those proportions in PKI placentas were similar to the control from E12.5 to E16.5.

After hematoxylin and eosin (H&E) staining, evident morphological abnormalities could be observed in MKI and HOMO labyrinth (Fig. 2d), whereas the maternal decidua and junctional zone didn't show obvious abnormality (Supplementary Fig. 2). MKI and HOMO mutants exhibited obviously enlarged vascular space and reduced vascular branches in the labyrinth from E12.5, compared to WT and PKI placentas. As early as E10.5, *Gtl2* polyA knock-in labyrinthine vasculature didn't show the difference, which may make sure the development of the embryos during the period of time (Supplementary Fig. 3a). The placentas of MKI and HOMO dead embryos at E14.5 and E16.5 also showed poor vasculature, however, we couldn't distinguish the abnormality caused by *Gtl2* polyA knock-in or by the death of embryos (Supplementary Fig. 3b).

The vascular network in the labyrinth consists of maternal blood sinus surrounded by syncytiotrophoblasts and fetal capillaries composed of fetal capillary endothelial cells[1]. We visualized the formation of fetal vasculature by performing immunohistochemistry (IHC) of endothelial marker CD31, which is broadly used to mark endothelial cells in the labyrinth[35]. The results revealed the endothelial cells distributed abnormally, some fetal capillaries dilated well, but most didn't dilate enough and poorly developed in MKI and HOMO labyrinth (Fig. 2e). The abnormal distribution of CD31 began to appear from E12.5 (Supplementary Fig. 3c–e). At E10.5, the MKI and HOMO fetal capillaries dilated well. *Dlk1* has been reported as a marker gene in fetal endothelial cells[36] and expresses restrictedly to the sites of branching morphogenesis[24]. DLK1 was detected to be uniformly expressed in the fetal endothelium, WT and PKI labyrinth formed normally tubular vessel structures. Instead, massive DLK1 clustered in MKI and HOMO labyrinth, and the fetal capillaries didn't branch well (Fig. 2f).

The maternal and fetal blood spaces (Fbs) can be easily distinguished from each other because the Fbs contain bigger nucleated embryonic erythrocytes and maternal blood spaces (Mbs) are filled with smaller non-nucleated mature erythrocytes[37]. Combined with the results of CD31 IHC,

we colored the Fbs with blue and Mbs with red (Fig. 2g). MKI and HOMO placentas had smaller Fbs and larger Mbs than those in WT and PKI at E14.5 (Fig. 2h, i).

As a summary, MKI and HOMO placentas show striking vasculature defect in the labyrinth. The narrow Fbs in MKI and HOMO maybe result in

the accumulation of fetal erythrocytes within them, leading to fetal blood vessels being clogged. This abnormality maybe affects the exchange functions of fetal capillaries ultimately.

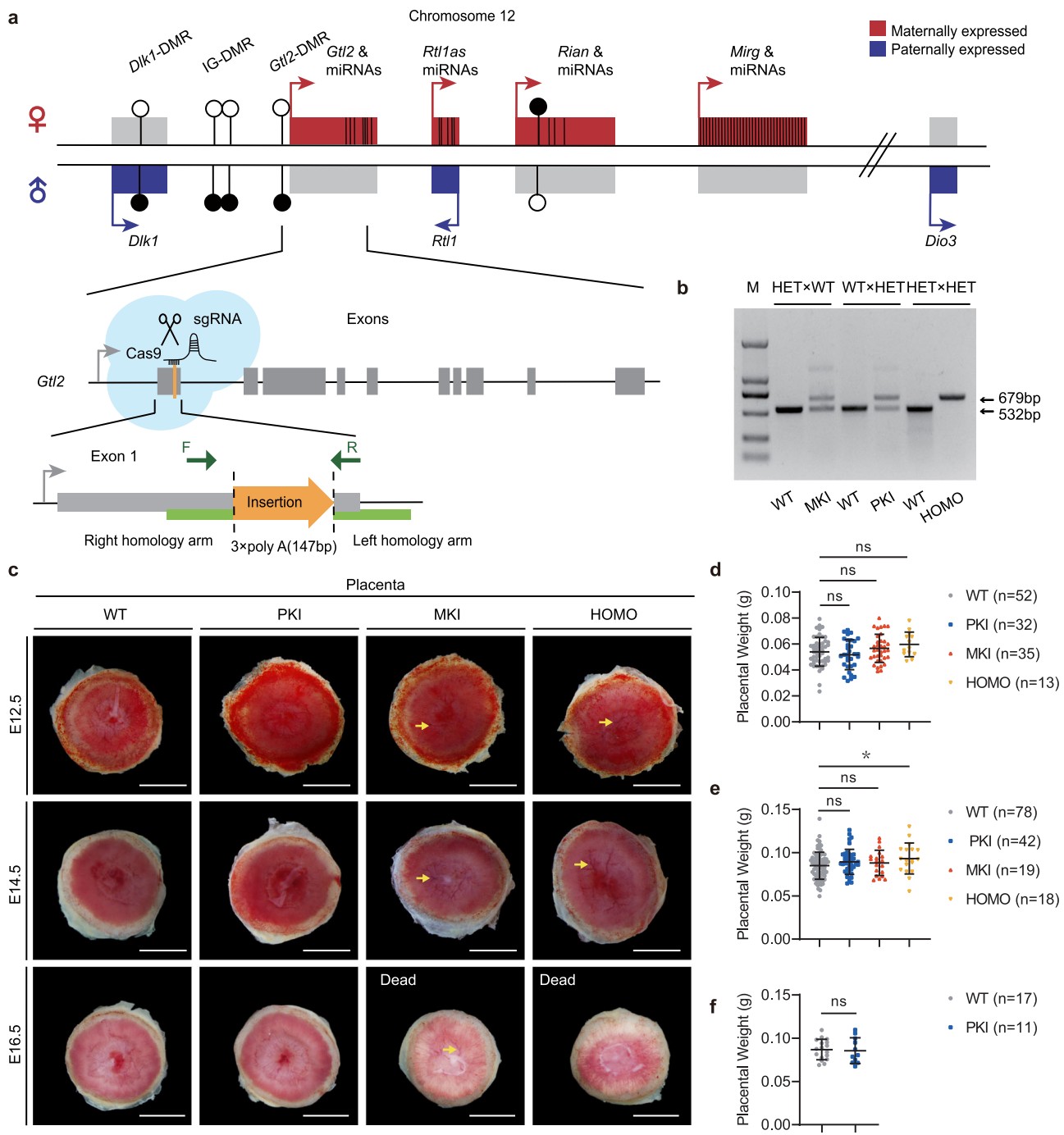

**Fig. 1 | Establish *Gtl2* polyA knock-in mouse model and analyze placental weights. a** Schematic representation of the *Dlk1-Dio3* imprinted domain and 3x polyA insertion position in *Gtl2* gene locus. Genes are shown as rectangles with their allelic expression state, maternally expressed genes are symbolized in red, paternally expressed genes in blue and not expressed genes in light gray. Arrows depict their transcription directions. The black vertical lines represent miRNAs. Differential methylation in the *Dlk1-Dio3* domain are shown with filled circles representing methylated allele and hollow circles representing unmethylated allele. Dark gray rectangles represent exons of *Gtl2*. The inserted sequence is in orange shape, and the green rectangles on both sides are the left and right homology arms separately. The positions of the primers for genotyping are indicated by the dark green arrows.

**b** Three mouse mating ways for getting the placentas and PCR confirmation of appropriate polyA knock-in. The wild type is 532 bp, and after right insertion the type is 679 bp. M: DL2000 DNA ladder (bp). **c** Gross phenotypes of WT and *Gtl2* polyA knock-in placentas at E12.5, E14.5 and E16.5. Scale bars: 2 mm. The yellow arrows indicate thicker and more obvious vessels in the labyrinth sides in MKI and HOMO placentas. **d–f** Comparison among WT and *Gtl2* polyA knock-in placental weights at E12.5 (WT $n = 52$, PKI $n = 32$, MKI $n = 35$, HOMO $n = 13$), E14.5 (WT $n = 78$, PKI $n = 42$, MKI $n = 19$, HOMO $n = 18$) and E16.5 (WT $n = 17$, PKI $n = 11$). The mean weight ± SD of each genotype is plotted. Studen *t*-test is used to analyze the *p*-value. *$p < 0.05$.

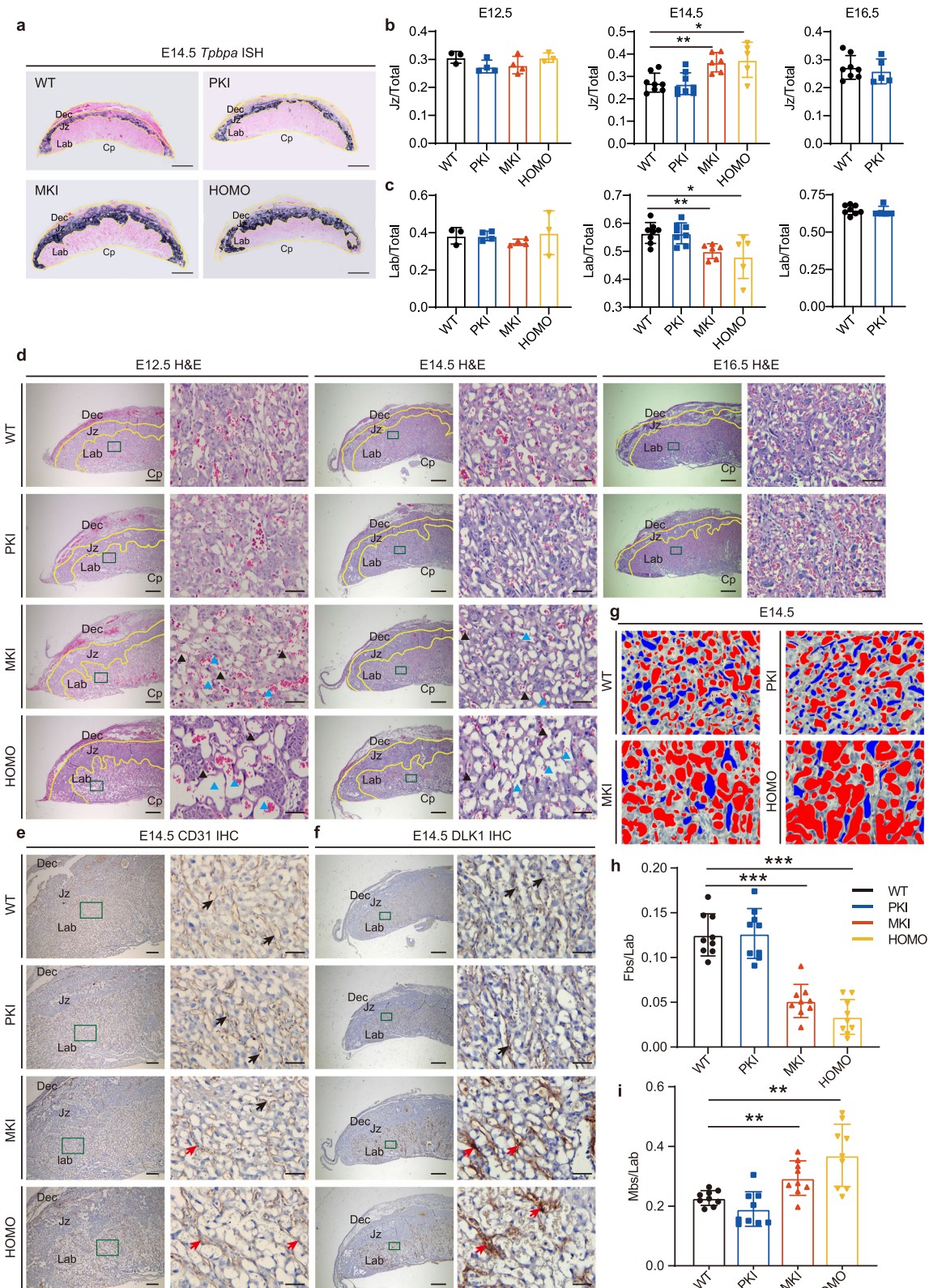

### Transcriptome analysis of *Gtl2* polyA knock-in placentas

To better understand the transcriptomic changes and molecular events in the *Gtl2* polyA knock-in placentas, we generated the transcriptome sequencing data from the placentas at E12.5.

We identified the differentially expressed mRNAs (DEGs) in PKI, MKI and HOMO placentas separately compared to WT. There were 1256 DEGs (up: 312, down: 944) in PKI, 572 DEGs (up: 314, down: 258) in MKI and 1081 DEGs (up: 749, down: 332) in HOMO (Fig. 3a–c). We found that two protein-coding genes located in *Dlk1-Dio3* domain, *Dlk1* and *Rtl1* were upregulated in MKI and HOMO. *Rtl1* was reported to play a crucial role in placental angiogenesis[38]. Some DEGs with significant changes and high expression levels were verified by qRT-PCR in Supplementary Fig. 4a–c.

**Fig. 2 | *Gtl2* polyA knock-in alters placental architecture and leads to vasculature defect in the labyrinth. a** In situ hybridization of *Tpbpa* on E14.5 placenta to distinguish the three-layer structure of the mature placentas. *Tpbpa*, which is the marker gene of spongiotrophoblast can mark the junctional zone of the placenta. Yellow lines distinguish three layers of placentas. Dec: decidua, Jz: junctional zone, Lab: labyrinth, Cp: chorionic plate. Scale bars: 1000 µm. **b**, **c** Proportions of the labyrinth area and junctional zone area compared to the total placental area at E12.5 (WT $n = 3$, PKI $n = 4$, MKI $n = 4$, HOMO $n = 3$), E14.5 (WT $n = 8$, PKI $n = 8$, MKI $n = 6$, HOMO $n = 5$), and E16.5 (WT $n = 8$, PKI $n = 5$). The mean proportion ± SD of each genotype is plotted. Student *t*-test is used to analyze the *p* values. *$p < 0.05$, **$p < 0.01$. **d** H&E staining of placentas at E12.5, E14.5, and E16.5. Dark green boxes show high-magnification images of labyrinth area (right). Blue triangles: maternal blood space (Mbs), black triangles: fetal blood space (Fbs). Scale bars of the first, third and fifth columns: 500 µm, scale bars of the second, fourth and sixth columns: 50 µm. **e** CD31 IHC of E14.5 placentas. CD31 is the marker of vascular endothelial cells. Dark green boxes show high-magnification images of labyrinth areas (right). Black arrows indicate CD31 express normally in the fetal vascular endothelial cells. Red arrows indicate aggregation distribution of CD31 in MKI and HOMO abnormal fetal vascular endothelial cells. Scale bars of the first column: 500µm, scale bars of the second column: 50 µm. **f** DLK1 IHC of E14.5 placentas. Dark green boxes show high-magnification images of labyrinth areas (right). Black arrows indicate DLK1 express normally in the fetal vascular endothelial cells, red arrows indicate aggregation distribution and increased expression of DLK1 in MKI and HOMO abnormal fetal vascular endothelial cells. Scale bars of the first column: 500 µm, scale bars of the second column: 50 µm. **g** According to the images of CD31 IHC of E14.5 placentas, the Fbs (blue) are surrounded by CD31, and filled with larger nucleated erythrocytes, and the Mbs (red) are not surrounded by CD31 and filled with smaller no nucleus erythrocytes. **h**, **i** The proportions of Fbs and Mbs compare to labyrinth at E14.5, respectively. Images of 3 microscopic views of one placental labyrinth are calculated to quantify the vascular area using ImageJ software. Numbers of placentas: WT $n = 3$, PKI $n = 3$, MKI $n = 3$, HOMO $n = 3$. The mean proportion ± SD of each genotype is plotted. Student *t*-test is used to analyze the *p* values. *$p < 0.05$, **$p < 0.01$, ***$p < 0.001$.

Venn diagrams were given to depict the overlapping upregulated or downregulated DEGs in different genetic types (Fig. 3d, e).

To better understand the functions of DEGs, we next did KEGG pathway and GO enrichment analysis, utilizing upregulated and downregulated DEGs in *Gtl2* polyA knock-in placentas, respectively (Supplementary Figs. 5–6). MKI and HOMO placentas had more upregulated DEGs, which might be responsible for the abnormal phenotypes. So, we focused on the KEGG (Fig. 3f) and GO (Supplementary Fig. 7) terms enriched by upregulated DEGs, and those were significant in MKI and HOMO, but not significant in PKI. Among these KEGG terms, 'Cytokine-cytokine receptor interaction' is key in luminal epithelium proliferation, trophoblast development, and uNK maturation during pregnancy[39]. 'ECM-receptor interaction' is involved in cell migration and vessel formation in placentation[40], especially in trophoblast giant cell migration[41]. Trophoblast giant cells lie at the edge of junctional zone, next to the maternal decidua in mouse. That may be related to the expansive junctional zone in MKI and HOMO placentas.

Since MKI and HOMO exhibited the same phenotypes, we next focused on the overlapped DEGs in MKI and HOMO placentas. The overlapped upregulated DEGs were more than the overlapped downregulated DEGs (91 vs 40) (Fig. 3d, e). Two overlapped upregulated DEGs were verified by qRT-PCR (Supplementary Fig. 4d). Among the GO terms enriched by the overlapped upregulated DEGs (Fig. 3g), 'Regulation Of Mitotic Spindle Checkpoint' functions in the differentiation of placental trophoblast stem cells to trophoblast giant cells[42]. Due to placental vasculature has a branching structure and the endothelial cells come from extra-embryonic mesodermal cells-derived allantois[1], 'Regulation Of Morphogenesis Of A Branching Structure', 'Mesodermal Cell Fate Commitment' are likely related to the MKI and HOMO phenotypes. The overlapped upregulated DEGs enriched in the KEGG terms (Fig. 3h) about the development and the functions of the placenta, like 'ECM-receptor interaction' and 'Hematopoietic cell lineage'. The abnormal fetal capillary endothelial cells in MKI and HOMO placentas may be responsible for 'Hematopoietic cell lineage', because a CD44$^+$ subpopulation of placental endothelial cells exhibits hemogenic potential[43].

### *Gtl2* polyA knock-in results in dysregulation of placental genes in the *Dlk1-Dio3* domain

We performed transcriptomics analysis combined with qRT-PCR at E12.5 to determine the effect of *Gtl2* polyA knock-in on placental gene expression of chr12 in detail.

The expression of all coding genes in chr12 were shown in lines in Fig. 4a. Most genes expressed stably in three kinds of *Gtl2* polyA knock-in placentas compared to their WT control, except *Dlk1* and *Rtl1*, located in the *Dlk1-Dio3* domain, which increased significantly in MKI and HOMO placentas. However, the expression of another PEG in that domain, *Dio3* had no changes in any kinds of *Gtl2* polyA knock-in placentas (Fig. 4b). In agreement with RNA-seq data, qRT-PCR showed that the expression of *Dlk1* in

MKI and HOMO placentas almost were twice as those in WT and PKI placentas, and *Rtl1* in MKI and HOMO placentas expressed almost 10-20 times as those in WT and PKI ones (Fig. 4c). In WT placentas, *Dlk1* and *Rtl1* only express from the paternal allele. We hypothesized those two genes may express from both alleles in MKI and HOMO placentas. *Dio3* didn't change expression in MKI and HOMO placentas, maybe because *Dio3* is not imprinted completely and biallelic expresses in the WT placenta[16].

All the lncRNAs expression in chr12 was shown in lines in Fig. 4d. The big changing lncRNAs located in the *Dlk1-Dio3* domain were annotated with gray shades. Contrary to coding genes, three main MEGs (*Gtl2*, *Rian*, and *Mirg*) had significant reduction in MKI and HOMO mutants (Fig. 4e). The qRT-PCR analysis confirmed that result. *Gtl2*, *Rian*, and *Mirg* hardly had any expression in MKI and HOMO ones, while their expression had not been affected in PKI by the transcription termination signals (Fig. 4f).

The *Dlk1-Dio3* domain harbors the biggest miRNA cluster in mice[44], and almost all the miRNAs are expressed from the maternally inherited allele as we know. Also, the strong changing miRNAs in chr12 were mostly concentrated in the *Dlk1-Dio3* domain. These miRNAs had striking reduction in expression in MKI and HOMO placentas, but not in PKI placentas (Fig. 4g). Furthermore, we categorized those miRNAs according to their locus in the genome with lncRNAs (*Gtl2*, *Rtl1as*, *Rian*, *Mirg*) and then showed the results in different color shades. As shown in Fig. 4h, small RNA-seq had detected 49 miRNAs originating from the *Dlk1-Dio3* domain, all the miRNAs reduced expression in MKI and HOMO placentas. While *miR-345* and *miR-1247* located separately before and after *Dlk1-Dio3* domain did not exhibit that expression trend. We selected several miRNAs (*miR-337*, *miR-127*, *miR-370*, *miR-381* and *miR-541*) in the domain that have high expression levels from each category, and then confirmed downregulation by qRT-PCR (Fig. 4i).

Summing up, *Gtl2* polyA knock-in mainly influences the *Dlk1-Dio3* domain in chr12. WT and PKI placentas have a same expression pattern, without big changes in PEGs or MEGs. While MKI and HOMO placentas also have a same expression pattern, PEGs expression increase significantly, and MEGs expression are almost silenced. Maternal expressed lncRNAs and miRNAs have the same silenced expression pattern because most maternal noncoding RNAs are expressed as part of a large polycistronic transcription unit starting at *Gtl2*[28], and they are regulated by polyA signal.

### *Gtl2* polyA knock-in results in dysregulation of genes in the *Dlk1-Dio3* domain in the labyrinth

To explore the underlying molecular mechanism of *Dlk1-Dio3* imprinted domain in *Gtl2* polyA knock-in impaired placental vasculature maintenance, we visualized and compared the spatial expression patterns of *Dlk1*, *Gtl2*, *Rtl1*, *miR-127* and *Rian* in the placental labyrinth of different genotypes by in situ hybridization at E12.5.

Endothelial cell specific expression of *Dlk1* were not been affected in PKI vasculature, but the transcripts were significantly upregulated in MKI

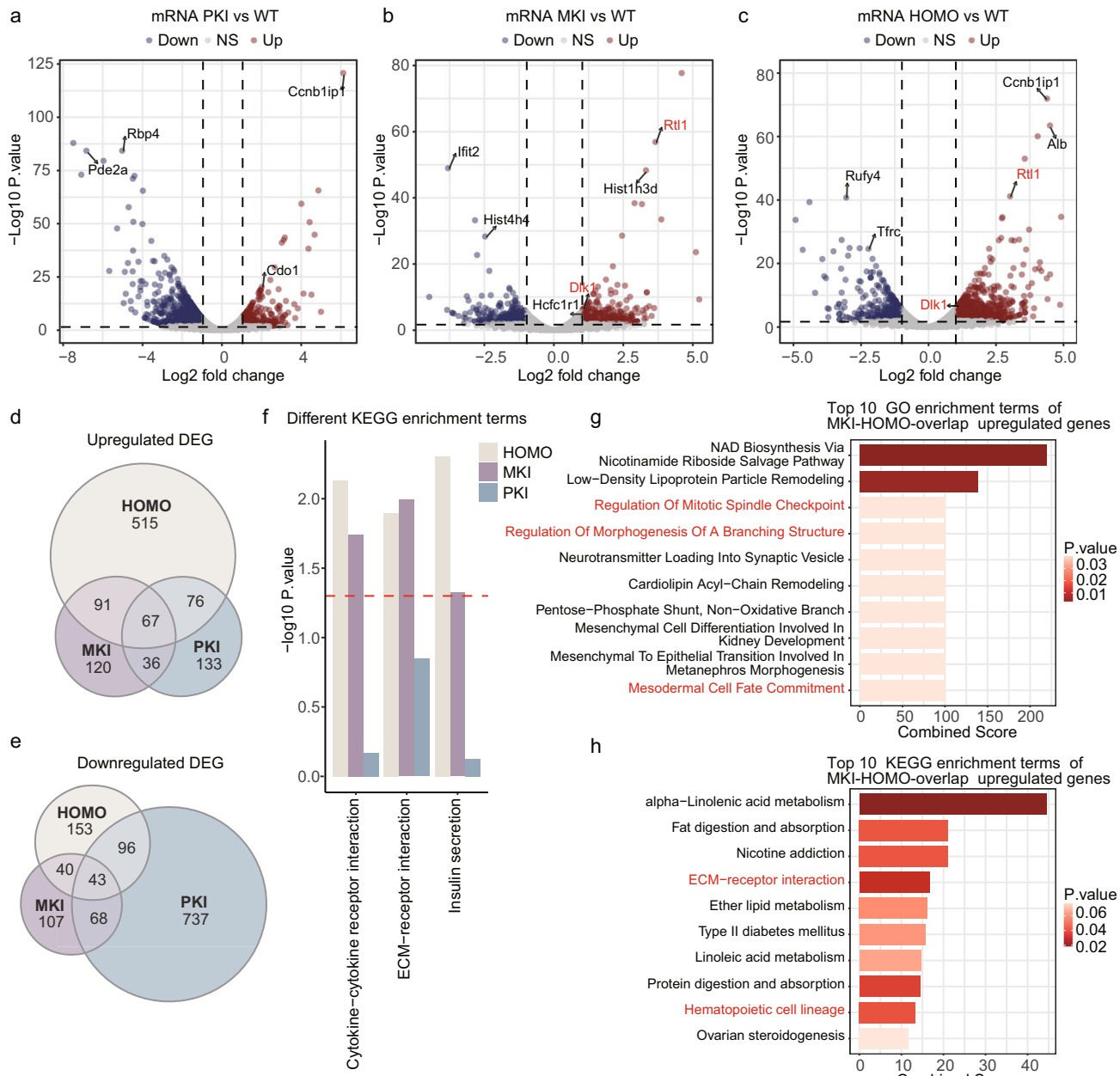

**Fig. 3 | Differential expression of mRNAs in *Gtl2* polyA knock-in placentas at E12.5. a–c** Volcano plots showing differentially expressed mRNAs (DEGs) in PKI, MKI, and HOMO compared to WT. The DEGs labeled in red are located in *Dlk1-Dio3* domain. The DEGs with significant changes and labeled in black are verified by qRT-PCR. **d, e** Venn diagrams of upregulated and downregulated DEGs in PKI, MKI and HOMO placentas, respectively. **f** The KEGG enrichment terms of upregulated DEGs are significant in MKI and HOMO, and not significant in PKI. The red dashed line represents the *p* is 0.05. **g** The top 10 GO enrichment terms of MKI and HOMO overlapped upregulated DEGs. Terms in red related to MKI and HOMO phenotypes. **h** The top 10 KEGG enrichment terms of MKI and HOMO overlapped upregulated DEGs. Terms in red related to MKI and HOMO phenotypes.

and HOMO endothelial cells without difference in expression position (Fig. 5a). The spatiotemporal expression patterns of *Dlk1* transcripts were consistent with those of DLK1 protein in *Gtl2* polyA knock-in placentas (Fig. 2e). In WT labyrinth vasculature, *Gtl2* expression signals were detected in dots around the fetal capillaries that contain bigger immature erythrocytes, just as the previous research reported[45]. PKI didn't change the position and intensity of *Gtl2* expression, and correspondingly there were no differences in phenotype in PKI labyrinth vasculature. In MKI and HOMO labyrinth, most signals disappeared, but occasionally several expression signals could be seen. PolyA signal knock-in could inhibit the transcription of *Gtl2*, but not completely (Fig. 5b). *Rtl1* and its regulator, *miR-127* expression concentrated in WT and PKI vasculature, but without strong signals. In MKI and HOMO placentas, *Rtl1* upregulated strikingly and *miR-127* hardly expressed (Fig.

5c, d). *Rian*[46] expressed around the fetal capillaries in WT and PKI vasculature, while it was hardly to see expression in MKI and HOMO ones, just like *Gtl2* (Fig. 5e).

MKI and HOMO showed low-level residual expression of MEGs, presumably reflecting read-through of the polyA signal by RNA polymerase. Generally, genes in *Dlk1-Dio3* domain like *Dlk1, Gtl2, Rtl1, miR-127 and Rian*, which dysregulated in MKI and HOMO labyrinth might directly cause placental vasculature abnormality.

### *Gtl2* polyA knock-in induces *Dlk1* loss imprinting
We observed *Dlk1* and *Rtl1* expression increased doubly and several times respectively in MKI and HOMO placentas. They might lose their imprinting and further activation might come from the normally repressed maternal

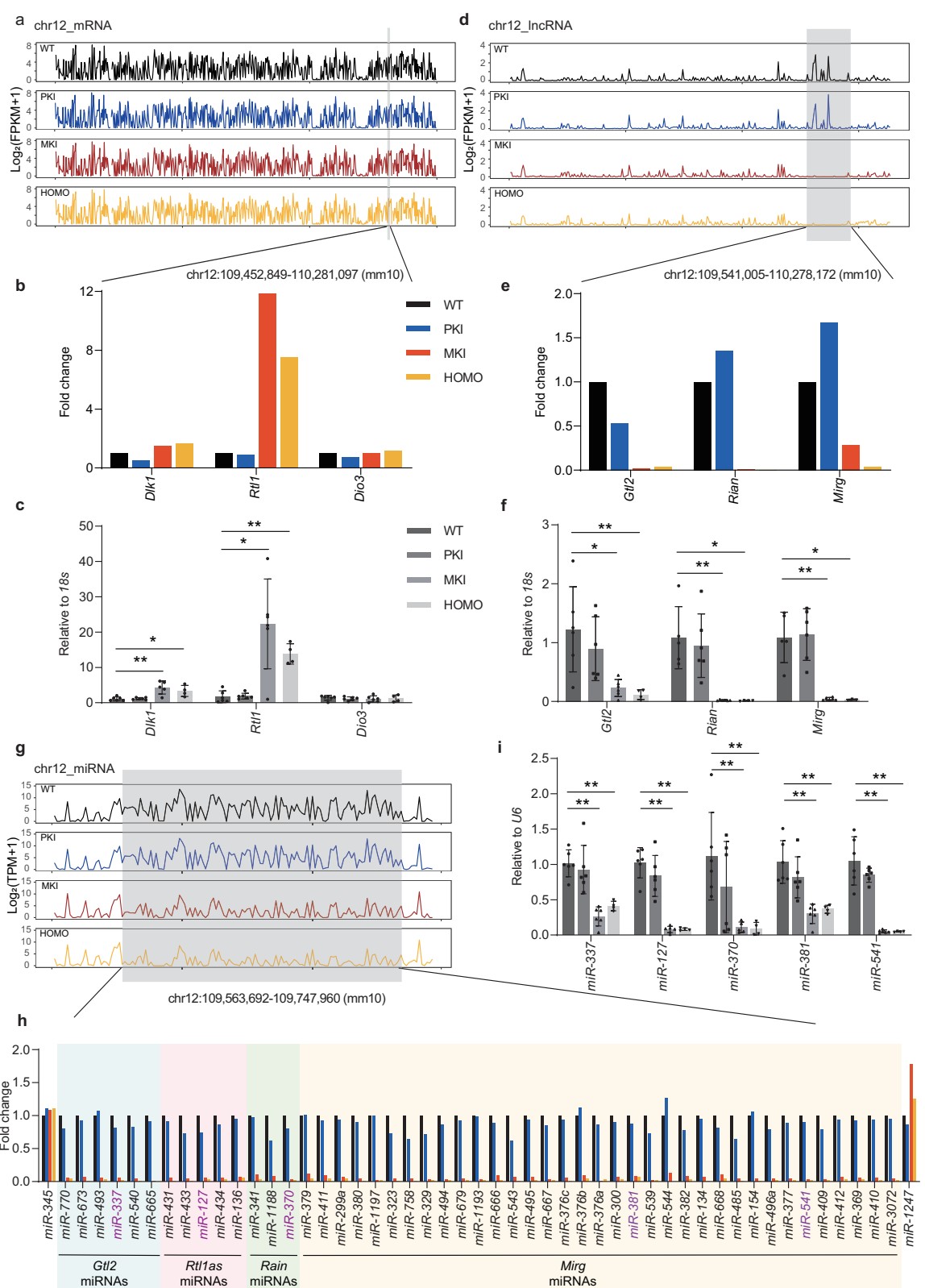

allele. We further proved our hypothesis using a single nucleotide polymorphism (SNP) of *Dlk1* between DBA/2 J (DBA) (A-T) and C57BL/6 N (B6) (G-C) mice. After amplifying the cDNA fragment containing the SNP and sequencing, *Dlk1* was indeed biallelically expressed in MKI and HOMO placentas at E12.5. *Dlk1* was normally expressed from paternal allele in PKI placentas (Fig. 6a). Strand-specific RT-PCR of *Dlk1* verified that result and

*Dlk1* had a similar expression level in both stands in MKI and HOMO (Supplementary Fig. 8a).

Upregulation of *Rtl1* was more striking in MKI and HOMO, much higher than twice that in WT. The results were consistent with the existing *Gtl2* knock-out mice, partly owing to loss of imprinting[9,27]. Overexpression of *Rtl1* also came from the absence of maternally expressed miRNAs. Several

**Fig. 4 | *Gtl2* polyA knock-in results in dysregulation of placental genes in the *Dlk1-Dio3* imprinted domain at E12.5. a** RNA-seq demonstrates mRNA expression in *Gtl2* polyA knock-in and WT placentas overall chr12. *Dlk1-Dio3* imprinted domain is annotated in gray shade. **b** Compare gene expression of *Dlk1, Rtl1* and *Dio3* in the *Dlk1-Dio3* domain among *Gtl2* polyA knock-in and WT placentas using RNA-seq data. **c** qRT-PCR analysis confirm the expression of *Dlk1, Rtl1* and *Dio3* to *18 s* in *Gtl2* polyA knock-in placentas. **d** RNA-seq demonstrates lncRNA expression in *Gtl2* polyA knock-in and WT placentas overall chr12. *Dlk1-Dio3* domain is annotated in gray shade. **e** Compared gene expression of *Gtl2, Rian,* and *Mirg* in the *Dlk1-Dio3* domain among *Gtl2* polyA knock-in and WT placentas using RNA-seq data. **f** qRT-PCR analysis confirm the expression of *Gtl2, Rian,* and *Mirg* to *18 s* in *Gtl2* polyA knock-in placentas. **g** Small RNA-seq demonstrates miRNA expression in *Gtl2* polyA knock-in and WT placentas overall chr12. *Dlk1-Dio3* domain is annotated in gray shade. **h** Compare expression of miRNAs in *Dlk1-Dio3* domain detected by small RNA-seq in four genotype placentas. Seven miRNAs shown in light blue shade are located in *Gtl2* locus, five miRNAs shown in light pink shade are located in *Rtl1* antisense locus, three miRNAs shown in light green shade are located in *Rian* locus, and twenty-six miRNAs showed in light yellow shade are located in *Mirg* locus. The miRNAs shown in purple have been detected by qRT-PCR. **i** qRT-PCR analysis confirm the expression of *miR-337, miR-127, miR-370, miR-381* and *miR-541*, which we select from each locus in *Dlk1-Dio3* domain whose expression abolish in MKI and HOMO placentas but are normal in PKI placentas. In (**c, f, i**), biological duplication: WT *n* = 6, PKI *n* = 6, MKI *n* = 6, HOMO *n* = 4. The mean expression ± SD of each genotype is plotted. Student *t*-test is used to analyze the *p* values. *$p < 0.05$, **$p < 0.01$. In (**b, c, e, f, h, i**), gene expression levels in WT are arbitrarily set to 1.

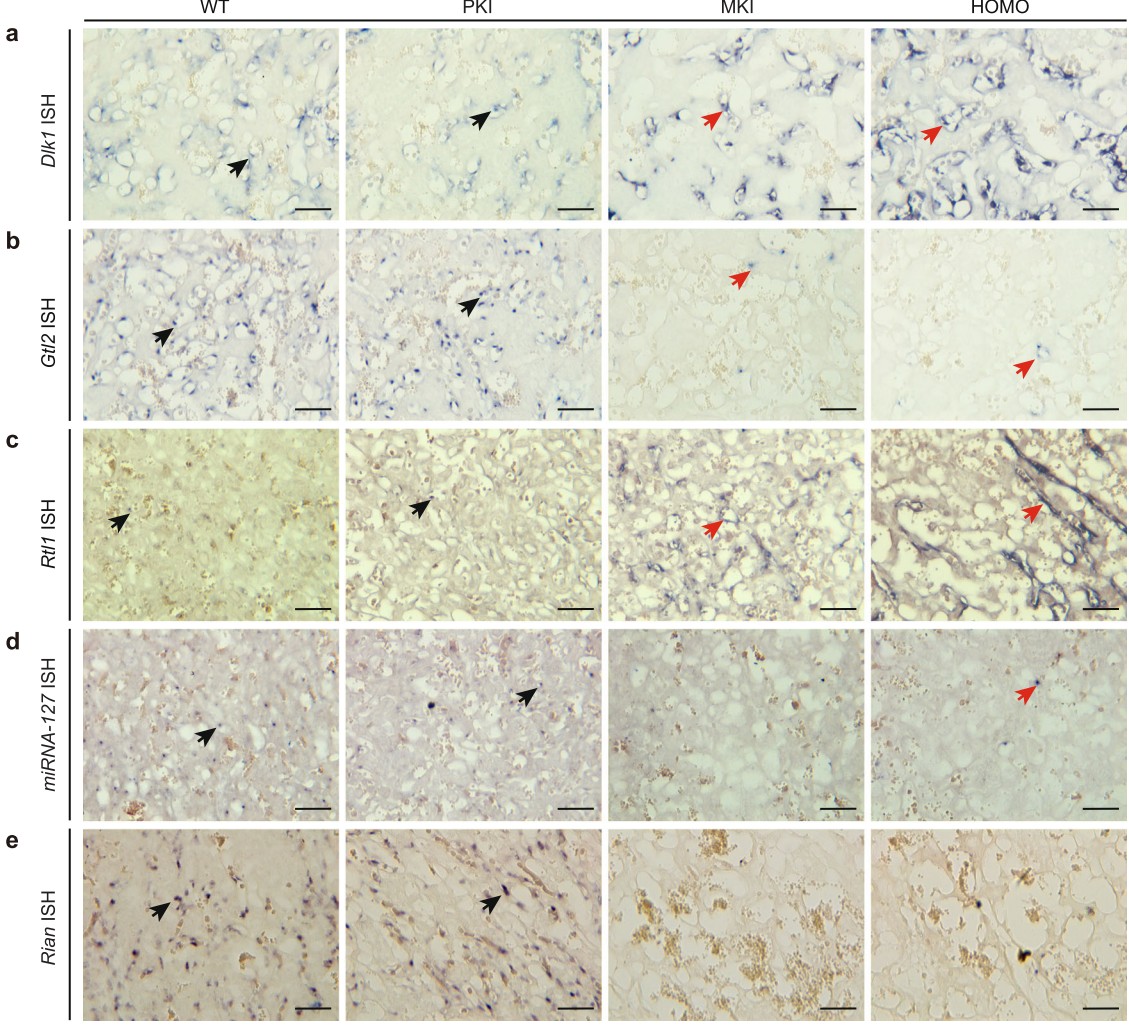

**Fig. 5 | Dysregulation of genes in the *Dlk1-Dio3* imprinted domain in the labyrinth of *Gtl2* polyA knock-in placentas at E12.5. a–e** In situ hybridization results show the expression pattern of *Dlk1, Gtl2, Rtl1, miR-127* and *Rian* in the labyrinth of *Gtl2* polyA knock-in and WT placentas, separately. Black arrows indicate gene expression normally in WT and PKI vasculature and red arrows indicate upregulated or downregulated gene expression in MKI and HOMO vasculature. In (**a–e**), scale bars: 50 μm.

miRNAs (eg. *miR-127* and *miR-136*) coming from *Rtl1* antisense (*Rtl1as*) can directly interact with *Rtl1* transcripts and partly degrade *Rtl1* transcripts through the RNAi-mediated method[21]. In MKI and HOMO placentas, all the miRNAs from *Rtl1as* almost had abolished expression (Fig. 4h), leading to *Rtl1* being upregulated significantly instead of the double dose expected from loss of imprinting.

During development, *Gtl2* expression can overlap the maternal *Dlk1* gene locus, and prevent *Dlk1* activation on the maternal allele to control its imprinting[47]. Thus, the abolished expression of MEGs lead to PEGs (*Dlk1*

and *Rtl1*) lose imprinting in MKI and HOMO placentas. Since imprinting errors are always causal to some developmental disorders, imprinting status change of PEGs may be associated with placental defects directly in MKI and HOMO placentas.

### Analyze methylation status of DMRs in *Dlk1-Dio3* domain in the *Gtl2* polyA knock-in placentas

To investigate how *Gtl2* polyA knock-in affects the up and downstream gene expression and imprinting status, and whether it functions through

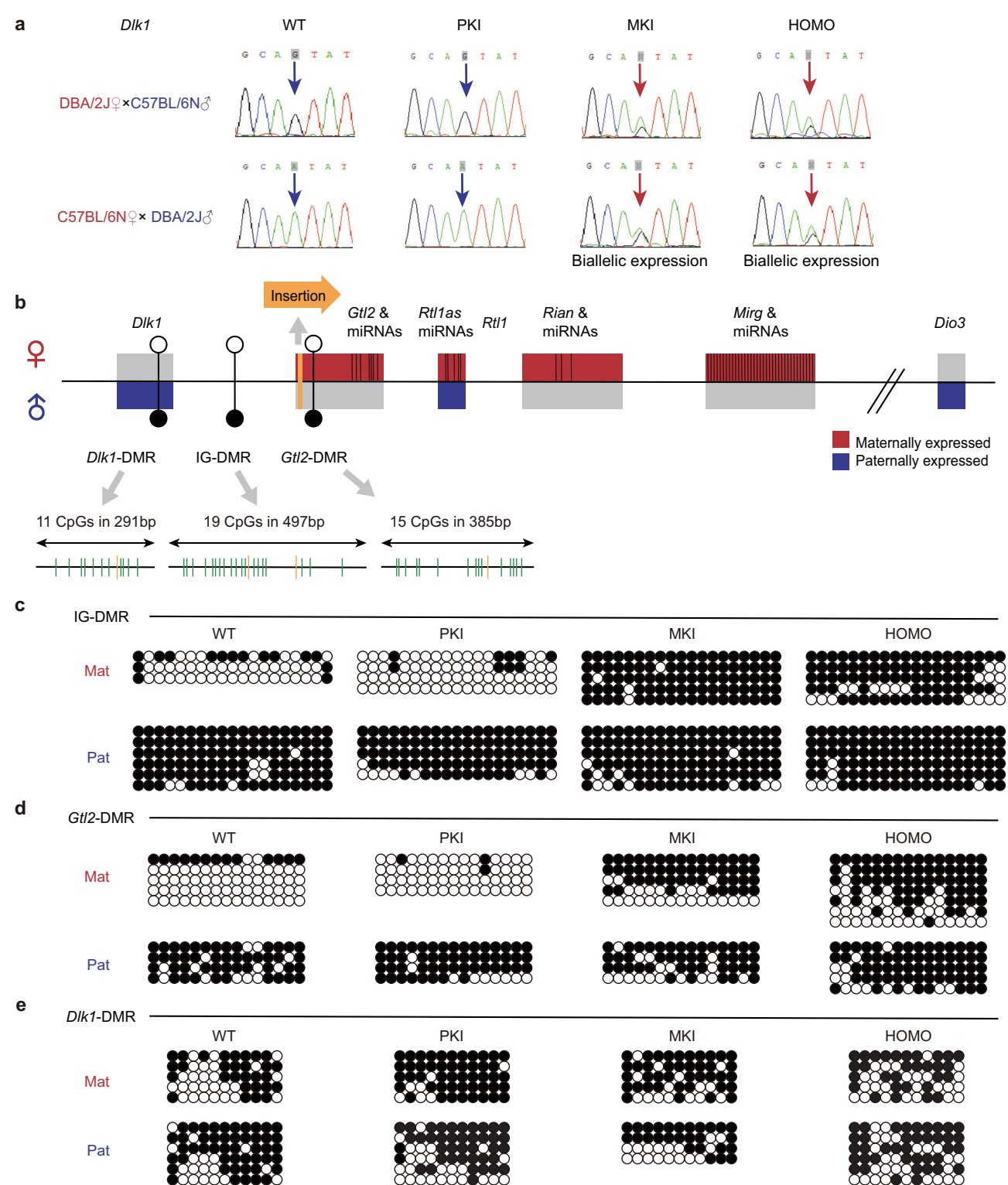

**Fig. 6 | Imprinting status analysis of *Dlk1* and DNA methylation analysis of DMRs in the *Dlk1-Dio3* imprinted domain in *Gtl2* polyA knock-in placentas at E12.5. a** Monoallelic expression of *Dlk1* in WT and PKI placentas and biallelic expression of *Dlk1* in MKI and HOMO placentas. Allelic-specific gene expression is determined by the SNP between C57BL/6 N (B6) and DBA/2 J (DBA) mice. Total RNA are extracted from *Gtl2* polyA knock-in placentas and their WT control. After purifying RNA and RT-PCR, PCR products containing the SNP for *Dlk1* are sequenced. Blue arrows indicate monoallelic *Dlk1* expression exclusively from the paternal allele and red arrows indicate detected biallelic *Dlk1* expression.

**b** Schematic representation of DMRs (IG-DMR, *Gtl2*-DMR, *Dlk1*-DMR) in *Dlk1-Dio3* domain. The inserted sequence is in orange arrow. Filled circles represent methylated DMRs and hollow circles represent unmethylated DMRs. Green vertical lines represent the CpG dinucleotides in each DMR, and the yellow vertical lines represent SNPs between B6 and DBA mice. **c–e** Methylation status of IG-DMR, *Gtl2*-DMR and *Dlk1*-DMR in *Gtl2* polyA knock-in and WT placentas separately. Each row represents one clone. SNPs are used to distinguish paternal allele clones from maternal allele clones. Filled and hollow circles indicate methylated and unmethylated CpG sites, respectively.

changing the methylation of DMRs in the placenta or not, we examined methylation status of DMRs at E12.5 by using the bisulfite-sequencing.

IG-DMR, located between *Dlk1* and *Gtl2*, functions as the main ICR in the *Dlk1-Dio3* domain. Its knock-out[8,48] or slight methylation status change[49] can change the imprinting status and secondary ICRs in the domain. We chose a 497 bp sequence of 19 CpG sites in IG-DMR to present it (Fig. 6b). This sequence contains two SNPs in DBA and B6 mice so that we can distinguish the paternal allele from maternal one[50]. As the result showed in Fig. 6c, IG-DMR was hypermethylated in the paternal allele and hypomethylated in the maternal allele in PKI placentas, consistent with WT. In MKI and HOMO placentas, IG-DMR became hypermethylated in both alleles.

*Gtl2*-DMR covers from the promoter to intron1 of *Gtl2*, as a secondary ICR for adjacent imprinted genes[51]. We detected the methylation of 15 CpG sites in a 385 bp sequence in *Gtl2*-DMR (Fig. 6b). This fragment is located in intron1 of *Gtl2* and behind the site of polyA knock-in. In WT placentas, *Gtl2*-DMR had parental origin-specific differential methylation status, with hypermethylation in the paternal allele and hypomethylation in the maternal allele. There was no change in methylation pattern observed in PKI placentas. Whereas, in MKI and HOMO placentas, the methylation levels increased, when the paternal alleles were hypermethylated, the maternal alleles were found to be hypermethylated, too (Fig. 6d).

*Dlk1*-DMR in the exon5 of *Dlk1* with paternal allele-specific methylation are found in many tissues, and biallelic methylation in placental tissue[10]. In three kinds of *Gtl2* polyA knock-in mice, *Dlk1*-DMR showed that hypermethylation on both alleles and the overall methylation levels had unaltered. So, the changes in expression and imprinting status of *Dlk1* were not related to *Dlk1*-DMR (Fig. 6e). The results of IG-DMR, *Gtl2*-DMR and *Dlk1*-DMR in other two biological replicates were provided in Supplementary Fig. 8b–d.

Those results indicate that *Gtl2* polyA knock-in regulates the gene expression and imprinting through leading methylation at IG-DMR and *Gtl2*-DMR in MKI and HOMO placentas. No changes in those DMRs make sure proper gene expression and imprinting in the *Dlk1-Dio3* domain in PKI placentas. *Gtl2*-DMR differential methylation in WT placentas initiates after the blastocyst stage and completes at E6.5[51]. The time of establishing *Gtl2*-DMR differential methylation pattern is later than the time of *Gtl2* imprinting expression[52], and we thought that the hypermethylated *Gtl2*-DMR functions to suppress MEGs expressed in MKI and HOMO placentas.

## Discussion

Despite *Dlk1-Dio3* imprinted domain is such important in prenatal and postnatal growth in both mouse and human[6,53], there are few studies focusing on MEGs in the developing placenta, which is critical to ensure embryonic development properly. Combined with *Gtl2* polyA knock-in model mice, our study explains that maternal RNA transcription plays important roles in maintaining vasculature in placental development. MKI and HOMO lead to expanded junctional zone and reduced labyrinth, with poor vasculature formation both in fetal and maternal blood spaces. MKI and HOMO dysregulate DNA methylation, imprinting status and then lead to gene expression dysregulated in the *Dlk1-Dio3* domain. PKI didn't show these changes (Fig. 7).

Here our model mice have shown great advantages. Unlike the existing *Gtl2* knock-out models with large-scale DNA sequence deletion[26,27], our *Gtl2* polyA knock-in mice are only inserted with a 147 bp DNA sequence, which eliminates the effect of varying degrees of the disruption of DNA elements therein, the missing of miRNAs and the change of chromatin structure. On the other hand, unlike conventional recombination technology in which the targeted region was replaced by a neomycin resistant gene (Neo) cassette, *Gtl2* polyA knock-in mice can avoid additional gene expression impact caused by the promoter of Neo. Maternal *Gtl2/Meg3*-DMR deletion (*Meg3*[Δ(1-4)/+]) mice in Wende Zhu's research exhibited dysregulating gene expression, loss-of-imprinting in the placentas and embryonic lethality, which are similar to our MKI and HOMO mice. *Meg3*[Δ(1-4)/+] placentas did not exhibit altered DNA methylation, which is quite different from our MKI

and HOMO placentas[9]. Different methods to establish model mice, operating on different genomic regions, and acquisition and deletion of sequences in those two model mice all likely lead to quite different effects on the genome and epigenetics (DNA methylation).

IG-DMR functions as the main ICR of *Dlk1-Dio3* imprinted domain, regulating imprinting status, secondary DMRs, and histone modifications[54]. *Gtl2* lncRNA can bind to PRC2[55] components (Ezh2, Eed, and Suz12) directly, and then they inhibit interaction of Ezh2/PRC2 at the IG-DMR locus and subsequent deposition of H3K27me3, which is associated with allelic DNA methylation. The presence of Ezh2/PRC2 in association with *Gtl2* lncRNA prevents Dnmt3s recruitment and subsequent de novo DNA methylation at IG-DMR, and ultimately keeps unmethylation in the maternal allele, and drives expression of the maternal *Gtl2-Rian-Mirg* locus[56]. The insertion of polyA signals prevented the transcription of *Gtl2* in MKI and HOMO placentas. In the absence of *Gtl2* lncRNA, maybe it is unable to prevent recruitment of Dnmt3s at the IG-DMR locus. Dnmt3s are then recruited to the IG-DMR and deposit de novo DNA methylation, leading to DNA methylation in the maternal IG-DMR. Hypomethylated maternal IG-DMR functions as an enhancer to activate MEGs transcription. When the hypomethylated maternal IG-DMR is methylated through CRISPR/Cas9 based targeted DNA methylation editing tools, the entire IG-DMR is hypermethylated and maternal *Gtl2* is repressed, *Dlk1* is expressed biallelically, and *Gtl2*-DMR gains methylation on the maternal allele[54]. Here, we think maternal RNA transcription ensures the right gene expression pattern, imprinting status and *Gtl2*-DMR methylation status through IG-DMR within the *Dlk1-Dio3* domain. Besides, *Gtl2* lncRNA can control the imprinting status of *Dlk1* directly. *Gtl2* lncRNA can partially retain in cis and overlap the maternal *Dlk1*, and prevent the deposit of H3K4me3 and H3K27ac, and then prevent *Dlk1* activation on the maternal allele[47]. The WT placenta displays biallelic methylation at *Dlk1*-DMR[10], and in PKI, MKI and HOMO placentas, the methylation status of *Dlk1*-DMR don't have altered. The DNA methylation status of the *Dlk1*-DMR may not be regulated by IG-DMR, and not function in regulating the expression and imprinting status of *Dlk1* in *Gtl2* polyA knock-in placentas.

Gene expression analysis concludes that MKI and HOMO placental abnormalities cannot be attributed to any single gene in the *Dlk1-Dio3* domain, because the whole domain is dysregulated, and multiple imprinted genes don't have the correct expression dosage. To figure out the regulation of each MEG in the domain, in future studies, different polyA knock-in model mice can be constructed through inserting transcriptional termination signal into *Rtl1as*, *Rian* and *Mirg*. Compared with different model mice, we can estimate the function of each MEG in histology, gene expression and epigenetics.

In the labyrinth, MKI and HOMO placentas cannot form fine, mesh-like structure of the fetal vasculature, where *Gtl2*, *Dlk1*, *Rian*, *Rtl1* and *miR-127* express dysregulated (Fig. 5). These genes were reported functional in both vascular endothelial cells and trophoblast cells. *Gtl2/Meg3* can protect endothelial functions by regulating the DNA damage response through P53 signaling[57]. *Gtl2/Meg3* deficiency induces cellular senescence of hepatic vascular endothelium and impairs glucose homeostasis and insulin resistance in obesity[58]. *Gtl2/Meg3* can regulate the proliferative and invasive capacities by activating the RAS-MAPK pathway[59], migration and apoptosis by Notch1 signal[60], and epithelial-mesenchymal transition by miR-210 of trophoblasts[61]. DLK1, as an atypical NOTCH ligand, has membrane-bound and secreted isoforms and functions in stimulating angiogenesis in the endothelial cells[62]. In the mouse placenta and embryo, *Dlk1* expresses in the sites of branching morphogenesis to mark the developing endothelium[24]. During placental development in human and mouse, *Rian/Meg8* is involved in regulating trophoblast functions[46]. *Rtl1/Peg10* was the first one in the domain proved essential for placental vasculature maintenance from mid-to late-gestation, even in horses[19,38].

Abnormality in the labyrinth will directly affect the substance transfer between maternal and fetal blood[2]. Placental vasculature defects may destroy placental hematopoietic niche, thereby affecting the fetal hematopoietic system, because placental vascular labyrinth provides a niche where

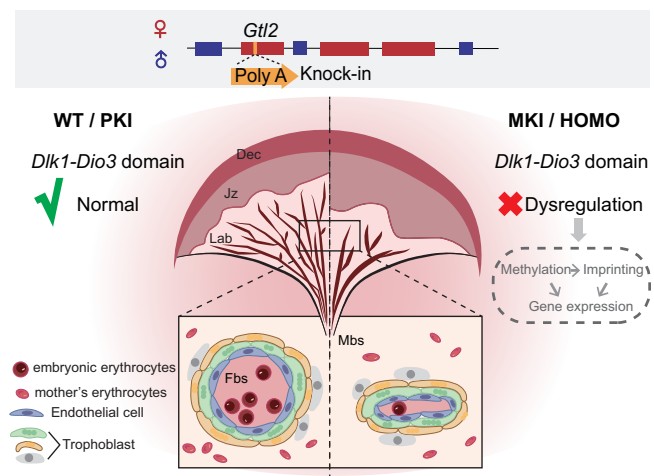

**Fig. 7 | A schematic illustrating the changes in *Gtl2* polyA knock-in placentas in this study.** PloyA signal is inserted into *Gtl2*. PKI placenta shows the similar phenotypes in placental morphology with WT, and *Dlk1-Dio3* imprinted domain works normally. MKI and HOMO placentas show the similar phenotypes, with expanded junctional zones and reduced labyrinth areas. Fetal blood spaces are reduced, and maternal blood spaces are larger in the labyrinth. Then, the DNA methylation and imprinted status are affected, leading to gene expression disregulated in the *Dlk1-Dio3* domain.

hematopoietic stem cells generate and expand[43,63]. In addition, junctional zone mainly consists of spongiotrophoblasts and glycogen trophoblasts. They function in remodeling maternal central arterial sinuses and produce many specialized products, like hormones, angiogenic factors and so on[1]. As another important part of the placental architecture, junctional zone expand in MKI and HOMO placentas without obvious difference in histology through H&E staining. We think the cell numbers in MKI and HOMO junctional zones might increase then lead to related functional changes. All the hypotheses should be verified by further studies. The histological phenotypes were not found in PKI placentas, which does not mean that there are no changes. In PKI placentas there exist the most DEGs and DEGs enriched in many KEGG and GO terms (Supplementary Figs. 5–6). Compared to MKI and HOMO placentas, the phenotypes of PKI placentas are more similar to WT.

It is undeniable that the abnormality of the placentas will directly affect *Gtl2* polyA knock-in embryonic development, however, we think that the current research about placental vasculature defects can't conclude that the abnormal placentas are the only cause of lethality in MKI and HOMO mice. Due to the presence of 'placenta-heart axis[64],' 'placenta-brain axis[65]' and so on, MKI and HOMO embryos may die from developmental disorders of multiple organs that exert reciprocal regulation[3,66]. We herein cannot attribute the lethality to one organ accurately because the model mice are global *Gtl2* polyA knock-in. Constructing conditional knockout mice or the tetraploid-chimera-based rescue of placental abnormality may point out whether the lethal effect organ is the placenta or not.

Our findings emphasize the maternal RNA transcription functions in regulating *Dlk1-Dio3* imprinted domain and makes sure placental vasculature in the mouse placenta. They will significantly facilitate deeper understanding of regulatory mechanisms in the *Dlk1-Dio3* imprinted domain and clear explaining of some pregnancy-related diseases associated with imprinted genes or placental malformations.

## Methods
### Ethics statement
Experimental procedures involving mice were performed in strict accordance with the Guide for the Care and Use of Laboratory Animals from the Harbin Institute of Technology (HIT) and approved by the Institutional Animal Care and Use Committee (IACUC-2023066) or Animal

Experimental Ethics Committee of HIT. We have complied with all relevant ethical regulations for animal use.

### Mice
The *Gtl2* polyA knock-in mice were generated by insertion of 3x polyA transcription termination signals into the intron 1 of *Gtl2*. The chromosomal position of insertion: mouse (GRCm38/mm10) chr12: after 109,541,067.

Based on the Easi-CRISPR technology, single guide RNA (sgRNA), single strand DNA donor (ssDNA donor), and Cas9 protein were prepared for microinjection. The sequences of sgRNA and ss DNA donor were described in Supplementary Table 2. After pronuclear injection to edit the zygotes of BDF1 (B6 × DBA) strain mice and surgically transplanting blastocysts into the uterus of surrogate mice, the founder with positive insert was successfully obtained.

The founder mouse was backcrossed into DBA and B6 genetic background, respectively, to gain *Gtl2* polyA knock-in mice with DBA background and B6 background. For imprinting analysis and methylation analysis, the *Gtl2* polyA knock-in mice in DBA background were used to mate with those in B6 background. All other experiments used *Gtl2* polyA knock-in mice in DBA background.

8 week-old DBA and B6 mice were procured from Charles River. All the mice were housed in a 12 h light–dark cycle with free access to food and water. Mouse mating was performed at night, and the next morning, when the copulation plug in the female was found, was designated as E0.5. The pregnant mice were sacrificed at the indicated times, and the placentas were collected after being dissected from the conceptus.

### Genotyping
Genomic DNA was isolated from the tails of postnatal mice or the yolk sac of embryos and was directly used as PCR template. PCR was performed using primer pairs to distinguish the WT and *Gtl2* polyA knock-in mutants. Band sizes: wild type, 532 bp; mutant, 679 bp. The primers were described in Supplementary Table 3.

### Histological analysis and image analysis
Placentas were fixed in 4% paraformaldehyde overnight, and then dehydrated in gradient ethanol to 100% ethanol, cleared in xylene substitute and infiltrated with paraffin. Then the tissues were embedded in paraffin wax. Tissue sections were cut into 5μm using a microtome, mounted on glass slides, and stained with H&E according to the standard procedures.

All slides were imaged using software ZEN (ZEISS) and image areas were analyzed using software ImageJ. The methods to distinguish the maternal and fetal blood spaces, and to calculate their relative proportions in the labyrinth have been previously described[67].

### In situ hybridization (ISH)
Amplify the part of cDNA of target genes as probes. The primers used were described in Supplementary Table 3. The amplifying cDNA fragment was inserted into pBlueScript II KS (+) vector. After single emyze digestion, the purified and linear cDNA fragment was used as RNA probe synthesis template. Then, digoxigenin (DIG)-labeled RNA probes were prepared according to the manufacturer's instructions by DIG RNA labeling kit (Roche). The RNA probes were stored at −80 °C until use.

The paraffin sections with 8 μm thickness were dewaxed and rehydrated. Digested the sections with 4 μg/ml proteinase K (Alphabiotech) at 37 °C for 15 min. Using Glycine terminated digestion reaction, the sections were put into acetylation solution (530 μl triethanolamine, 70 μl 37% HCl, 100 μl acetic acid, DEPC-treated water to 40 ml) for 15 min at room temperature. Sections were pre-hybridized in hybridization buffer (50% formamide, 5*SSC, 1*Denhardt' solution, 10 mg/ml yeast tRNA, 10 mg/ml salmon sperm DNA) at 65 °C for 1 h and were incubated with RNA probes (500 ng/ml in hybridization buffer) at 65 °C for overnight. Sections were washed with 0.2*SSC at 65 °C and were blocked with blocking buffer (10% skim milk powder in maleic acid buffer) for 1 h at room temperature and

were incubated at 4 °C overnight with alkaline phosphatase-conjugated anti-DIG antibody (1:3000 dilution, 54732420, Roche). Hybridization signals were detected with NBT/BCIP Stock Solution (Roche).

The sections for *Tpbpa* were stained with eosin finally.

## Immunohistochemistry (IHC)

The paraffin sections with 8 μm thickness were dewaxed and rehydrated. Antigen retrieval was performed by microwaving samples in citrate buffer at pH 6.0. After blocking with 3% bovine serum albumin (BSA, BioFroxx) in PBS for 1 h at room temperature, the sections were incubated with primary antibodies in blocking buffer overnight at 4 °C. Primary antibodies used were: rabbit monoclonal anti-CD31 (1:100 dilution, #77699, CST); rabbit monoclonal anti-DLK1 (1:200 dilution, ab210471, Abcam). After washing with PBS, the sections were incubated with HRP Goat Anti-Rabbit IgG (1:5000 dilution, AS014, ABclonal) in PBS for 1 h at 37 °C. Chromogenic detection was performed by DAB (ZSGB-BIO) staining. When the appropriate color appeared, the sections were stained with hematoxylin.

## RNA isolation and quantitative real-time RT-PCR

Total RNA was isolated from E12.5 placentas by using RNAiso (TakaRa) and was treated with DNase I (Thermo Scientific) to remove genomic DNA. Recombinant RNase Inhibitor (TakaRa) was used to protect RNA from degradation.

To detect mRNA and lncRNA expression, the RNA was used to synthesize cDNA by PrimeScript™ RT reagent Kit (TakaRa). Reverse transcription for miRNA was performed using Mir-X™ miRNA First-Strand Synthesis (TakaRa). Then, quantitative PCR (CFX96™ Real-Time System, Bio-Rad) was performed using TB Green® Premix Ex Taq™ II (Tli RNase H Plus).

*18s* was used as the internal reference gene for mRNA and lncRNA. *U6* was used as the internal reference gene for miRNA. All the primers used in qPCR were described in Supplementary Table 3. The relative expression was calculated by delta-delta CT formula and normalized to the internal reference gene.

## Biallelic expression of *Dlk1*

The biallelic expression of *Dlk1* was demonstrated using a SNP identified between DBA and B6 mice[8]. The cDNA fragment containing the SNP was amplified by PCR. After gel purification, the fragment was sequenced to identify the SNP. The primers were described in Supplementary Table 3.

## Methylation analysis

Genomic DNA was isolated by using the proteinase K/SDS and phenol-chloroform method from placentas with the maternal decidua removed to avoid the influence of mothers' DNA. The purified genomic DNA (500 ng) was treated with sodium bisulfite solution by using EZ DNA Methylation-Gold™ kit (Zymo Research). The resulting DNA was amplified by the primers for IG-DMR[50], *Gtl2*-DMR and *Dlk1*-DMR. All the primers' sequences were described in Supplementary Table 3.

The DNA fragments were separated electrophoretically on 1% agarose gels to purify the PCR products, and the bands were cut off and purified using phenol-chloroform method. The purified DNA was cloned into the pMD™19-T Vector Cloning Kit (TakaRa) and was sequenced (Comate Bioscience, China).

## RNA-seq data analysis

After verifying the quantity and quality of RNA using Nanodrop (Thermo Fisher Scientific) and RNA electrophoresis, respectively, the total RNA was provided to Bio Company (BMKGENE, China) for the whole transcriptome sequencing. The cDNA library was generated and sequenced on Illumina HiSeq sequencing platform. All reads were aligned to mouse GRCm38 genome using HISAT2[68], and genes were counted using StringTie[69]. For miRNA, the mapped reads were aligned with miRbase[70].

The default parameters of edgeR[71] were used, and DEGs were selected based on log2 fold-change > 1 and *p* < 0.01. Since there was no biological replication in this study, the biological coefficient of variation (BCV), which is the square-root of dispersion, was set to 0.1 according to the suggestion of the edgeR official manual. Functional enrichment analysis was conducted using Enrichr[72]. For Biological Process categories, we used Go terms, and other parameters were left at their default settings.

## Statistic and reproducibility

Statistical analyses were performed using GraphPad Prism 9.0.2. Data obtained from each experiment were expressed as the mean ± SD. Statistical significance between two groups was determined by two-tailed unpaired *t*-tests. $P < 0.05$ were accepted as statistically significant and were indicated as follows: $*P < 0.05$, $**P < 0.01$, $***P < 0.001$, ns not significant. The sample sizes were described in each figure legend.

## Reporting summary

Further information on research design is available in the Nature Portfolio Reporting Summary linked to this article.

## Data availability

The source data for graphs are available in Supplementary Data 1. RNA-seq data have been deposited in Gene Expression Omnibus database (GEO) under accession number GSE255900. All the other data generated in this study are included in the article and the additional files.

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

## Acknowledgements

This work was funded by National Natural Science Foundation of China [U20A20376, 61972116]; Applied Technology Research and Development Project of Heilongjiang [GA20C018]; Heilongjiang Provincial Natural Science Foundation (LH2019C038).

## Author contributions

Q.W. conceived the research and designed the experiment. X.Z. performed the experiments, analyzed the data, and wrote the paper. H.H., F.Z. and Y.Z edited the paper. H.R.Y. performed qRT-PCR experiment. X.T. performed genotype identification experiment. C.L. and J.S. analyzed the RNA-seq data. Z.W. and H.P.Y. provided the materials. T.W. and J.L. took the pictures. All authors reviewed the results and approved the final manuscript.

## Competing interests

The authors declare no competing interests.
