## [Peer Review File · Communications Biology]

Reviewers' comments:

Reviewer #1 (Remarks to the Author):

This article by Zhang et al generated a new mouse model in which PolyA termination signals were inserted into the Dlk1-Dio3 imprinted domain, which modulated expression of maternal RNAs in this domain. Notably, altered placental development and vascularity were evident in maternal knock-in or homozygous (maternal and paternal knock-in) embryos, but not paternal knock-in embryos. Overall, the mouse model is innovative and the data are comprehensive and interesting. Statistical analysis and methodological details appear to be appropriate. A few suggestions for improvement are listed below:

1. The authors suggest that the mature mouse placenta is not formed until embryonic day 12.5, but vascular invasion/expansion happens several days earlier. Is the Dlk1-Dio3 cluster needed for initial vascular expansion into the placenta, or are vessels in the MKI and HOMO placentas regressing?
2. Similar to the above question, in Figure 2 fetal and maternal blood spaces are quantified on day 14.5 in the four groups. Is there any change notable on day 12.5? This is important given that all embryos are viable on this day, whereas many have died by day 14.5.
3. Are there alterations in the trophoblast trilaminar structure in MKI and HOMO placentas?
4. In Figure 5, some of the images appear to be poor quality and hard to see (e.g., it is difficult to see what the arrows are pointing at in the panels showing miR-127 and Rian). Likewise, a few panels in Figure 5, and all the images in Figures S2 and S3 have a very yellow background and the images should be improved.

Reviewer #2 (Remarks to the Author):

In this manuscript, the authors focused on the Gtl2-Dio3 region, one of the paternally imprinted genes, and created mice in which polyA sequences were inserted into the Gtl2 gene to suppress the expression of downstream genes. Gene expression and DNA methylation were abnormal when the PolyA knock-in allele was the maternal allele (MKI) or homozygous (Homo), resulting in abnormal placenta formation and embryonic lethality. The developmental phenotype of the placenta is clear and interesting. Moreover, changes in gene expression and DNA methylation are evident around the knock-in region. However, the RNA sequencing analysis is inadequate, and the problem is that there is no causal link between polyA knock-in, methylation changes, gene expression changes, and placental abnormalities. Specific comments are noted below.

Major Comments

1. Line 181: Is there any experimental evidence showing the "clogging blood vessels"? Evidence from histology or alternatives should be provided.
2. Figure 3: RNA-seq should be analyzed in greater detail. For example, a detailed characterization of each cluster of DEGs should be included. It is strange why the authors did not focus on the commonly

changed genes between MKI and HOMO compared to WT. Which kind of GO term was enriched in these common DEGs? It may not be a good idea to analyze up- and down-regulated genes together, since the implications of these DEGs are different. The representative examples of differentially expressed genes/miRNA/lncRNAs should be shown in snapshots or bar charts. What is the sample size for RNA-seq analysis? The authors should analyze at least two biological replicates because the developmental stage should be taken into account. To call DEGs, add a p-value, not just fold changes. This dataset can help the authors explain why the junction zone is expanded in MKI and HOMO from E12.5 to E14.5.

3. Line 189-190: What is the possible explanation for more DEGs in PKI than MKI (882 vs 294), even though little phenotype is observed in PKI? According to Sup Fig 4A, several GO terms were significantly enriched in comparison to MKI. Does this result suggest the PKI has some gene expression changes independent of imprinting?

4. Line 407-409: "MKI and HOMO embryos with a global Gtl2 polyA knock-in died after approximately E12.5 accompanied by multiple phenotypes in different organs in parallel, including the placenta, liver, and heart, and so on." No phenotype information of the liver and other tissues is provided in the manuscript. An embryonic Gtl2 function could be revealed by the tetraploid-chimera-based rescue of placental abnormality.

5. Figure 6: Why are the IG-DMR and Gtl2-DMR hypermethylated in MKI and Homo mice? Would this indicate that Gtl2 3' ncRNAs play a role in imprint maintenance? In the maternal Meg3-DMR deletion mouse, loss-of-imprinting and embryonic lethality were observed, which is similar to the current manuscript (PMID: 31301299). However, Meg3-DMR deletion mice did not exhibit altered DNA methylation. It is important to analyze DNA methylation and gene expression at earlier time points (e.g., E6.5 ectoplacental cone and E10.5 early placenta) to gain a better understanding of the mechanism.

Minor Comments

6. Figure 1A-B: Gtl2 locus gene expression changes in Poly-A knock-in mice were not verified in Figure 1. While the authors confirm the silencing of MEGs in Gtl2 locus in MKI and HOMO placentas in Figure 4, conducting the phenotype analysis is logically unclear without confirming gene expression change. The reviewer suggests moving the RT-qPCR results from Figure 4c,f,i to Figure 1.

7. Line 169 and 173: Figure 2F appeared before Figure 2E. The figure order should be reorganized.

8. There is no consistency in the nomenclature of Gtl2 throughout the manuscript; Gtl2, Gtl2/Meg3, and Meg3 are used randomly. The official gene symbol is Meg3, although Gtl2 has historically been used widely. Therefore, this reviewer believes that using Gtl2/Meg3 rather than Gtl2 would be beneficial to readers.

9. Lines 433-434: Information about the strain is elusive. It seems likely that the authors generated KI mice with a B6 background first, then crossed them with DBA to produce BDF1 KI mice for allele-specific expression analysis. How about other experiments? Did authors always use mice with DBA backgrounds?

This part needs to be clarified.

Reviewer #3 (Remarks to the Author):

In this manuscript, Zhang et al. examined the role of maternal RNA transcription in the Dlk1-Dio3 imprinted domain in placental development. Taking advantage of the feature that the transcripts in the maternal allele in this domain are generated by polycistronic transcription from the Gtl2 promoter, they inserted a polyA cassette at the 3'-end of the Gtl2 promoter and generated a mouse model in which the expression of MEGs (Gtl2, Rian, Mirg and miRNAs) is downregulated, while the Gtl2-DMR sequence is retained and the genome context is minimally altered. They reported that maternal allele polyA knock-in (MKI) and both alleles polyA knock-in (HOMO) embryos died in mid-gestation. In MKI and HOMO placentas, they found that proportion of junctional zones and labyrinth areas were altered and that vascular formation in the labyrinth was abnormal. They also detected de-repression of PEGs (Dlk1 and Rtl1), as well as aberrant DNA methylation of the Gtl2-DMR and IG-DMR on the maternal allele in MKI and HOMO placentas. They did not observe these changes under paternal knock-in (PKI). They concluded that maternal RNA transcription in the Dlk1-Dio3 domain is essential for proper placentation.

The experiments were nicely done and the manuscript was clearly written. The results of this study, which were capable of demonstrating that maternal RNA transcription in the Dlk1-Dio3 domain is required for normal placentation without changing the genome sequence as much as possible, are significant for understanding the regulation of both ontogeny and genomic imprinting. However, due to the altered expression levels of multiple MEGs and PEGs in the domain, the molecular mechanisms of which gene products (proteins, lncRNAs and miRNAs) directly or indirectly regulate placental development have not been elucidated. As the authors discuss, it would be useful to generate mouse models with polyA cassettes inserted downstream of each MEG in future studies. Below I have listed several points, in which the manuscript needs to be improved.

1) Lines 98-101, Fig. 1A

Please indicate the chromosomal position (coordinate) where the polyA cassette was inserted. Sequence information of the donor DNA fragment used in CRISPR genome editing should also be provided.

2) Lines 133-136 and lines 407-409

The body weight, liver, and heart phenotypes of Gtl2 polyA knock-in embryos are mentioned, but no relevant data are presented. They should be presented.

3) Lines 165-173

Figure 2 panels presented out of order (Fig. 2E is explained after 2F).

4) Fig. 6A

Allelic-specific expression of the Dlk1 gene should be quantified (e.g. by pyrosequencing) for multiple

samples (biological replicates).

5) Fig. 6B

In the Dlk1-Dio3 domain map, the insertion site of the polyA signal and the regions analyzed by bisulfite sequencing should be illustrated to help better understanding.

6) Figs. 6C-E

Was DNA methylation analysis of each genotype by bisulfite sequencing conducted using a DNA sample extracted from only one placenta? To ensure reproducibility of data, it is important to employ biological replicates and the results for multiple samples should be presented.

7) Figs. 6C-E

Gtl2 polyA knock-in caused loss-of-imprinted DNA methylation (gain of methylation on the maternal KI allele) in Gtl2-DMR and IG-DMR, but not in Dlk1-DMR. Discussion on the molecular mechanism of this phenotype should be included.

8) Line 410

The "brian" should be corrected to "brain".

9) Supplementary Table 1

Please check that the primer sequences for DNA methylation analysis are correct. Some primers contain both "C" and "G" in the sequence, which is unusual for bisulfite sequencing primers.

Point-by-Point Response to Reviewers

Reviewers' comments:

*Responses to the reviewers' comments are highlighted in blue.

Reviewer #1 (Remarks to the Author):

This article by Zhang et al generated a new mouse model in which PolyA termination signals were inserted into the *Dlk1-Dio3* imprinted domain, which modulated expression of maternal RNAs in this domain. Notably, altered placental development and vascularity were evident in maternal knock-in or homozygous (maternal and paternal knock-in) embryos, but not paternal knock-in embryos. Overall, the mouse model is innovative and the data are comprehensive and interesting. Statistical analysis and methodological details appear to be appropriate. A few suggestions for improvement are listed below:

Thank you for your recognition of our work, we are so grateful for your valuable advice, contributing to the improvement of the manuscript.

1. The authors suggest that the mature mouse placenta is not formed until embryonic day 12.5, but vascular invasion/expansion happens several days earlier. Is the *Dlk1-Dio3* cluster needed for initial vascular expansion into the placenta, or are vessels in the MKI and HOMO placentas regressing?

Response:

Thank you for the problem. On the one hand, we think the maternal RNA transcriptions in *Dlk1-Dio3* domain affect the development of the placental vasculature rather than regressing. We have found the abnormal vasculature as early as E12.5 (Fig.2d and Supplementary Fig.3d-e). All the results of CD31 IHC at E12.5, E13.5 and E14.5 are from the placentas of alive embryos. (The survival rates of *Gtl2* polyA knock-in embryos are shown in Supplementary Table1.) Those prove the abnormal placental vasculature are not regressing due to the dying embryos.

On the other hand, it seems that the maternal RNA transcriptions don't affect the initial vascular expansion. After E10, the structure of the primitive choriovitelline ("yolk sac") placenta is beginning the transition to the more mature chorioallantoic ("labyrinthine") configuration, which is termed the "definitive" placenta once it develops fully around E12.5^[1]. So we have done the morphological analysis at E10.5, and don't observe vasculature abnormalities in MKI and HOMO (Supplementary Fig.3a and 3c). The fetal capillaries dilate and CD31 don't show aggregated distribution. All MKI and HOMO embryos are lived and begin to die after E12.5. The early placentas with normal morphology may ensure the development of the embryos during the period of time.

In brief, we think the maternal RNA transcriptions just affect the development of the placental vasculature at midgestation (after E12.5).

Lines in revised manuscript: 163-167

Revised text: 'MKI and HOMO mutants exhibited obviously enlarged vascular space and reduced vascular branches in the labyrinth from E12.5, compared to WT and PKI placentas. As early as E10.5, Gtl2 polyA knock-in labyrinthine vasculature didn't show the difference, which may make sure the development of the embryos during the period of time (Supplementary Fig.3a).'

Lines in revised manuscript: 177-179

Revised text: 'The abnormal distribution of CD31 began to appear from E12.5 (Supplementary Fig.3c-e). At E10.5, the MKI and HOMO fetal capillaries dilated well.'

Revised figure: Supplementary Fig.3a, 3c, 3d and 3e

2. Similar to the above question, in Figure 2 fetal and maternal blood spaces are quantified on day 14.5 in the four groups. Is there any change notable on day 12.5? This is important given that all embryos are viable on this day, whereas many have died by day 14.5.

Response:

The notable changes begin to be observed at E12.5 (Fig.2d). We have supplemented the results of CD31 IHC at E12.5 and E13.5 to prove that (Supplementary Fig.3d and 3e).

Lines in revised manuscript: 177-178

Revised text: 'The abnormal distribution of CD31 began to appear from E12.5 (Supplementary Fig.3c-e).'

Revised figure: Supplementary Fig.3d and 3e

3. Are there alterations in the trophoblast trilaminar structure in MKI and HOMO placentas?

Response:

Thank you for the good question, and it's the problem we need to solve next. In the labyrinth, the fetal vasculature is composed of fetal capillary endothelial cells and three surrounding layers of trophoblast cells: two syncytiotrophoblast (SynT-I and II) layers and one mononuclear sinusoidal trophoblast giant cell (s-TGC) layer. We have found such big change in the endothelial cells. We are curious if their adjacent trophoblast cells are abnormal, too. Currently, we cannot find the changes through the histological analysis. Our research group is going to combine single-cell RNA seq and visualization of their marker genes to study trophoblast cells in the future. Now we cannot get any conclusion about trophoblast cells.

4. In Figure 5, some of the images appear to be poor quality and hard to see (e.g., it is difficult to see what the arrows are pointing at in the panels showing miR-127 and Rian). Likewise, a few panels in Figure 5, and all the images in Figures S2 and S3 have a very yellow background and the images should be improved.

Response:

The old version of manuscript we uploaded has been compressed because the images are needed to be integrated into the article. In accordance with your helpful advice, we have improved all the images in the revised manuscript and uploaded them separately in raw format.

Because *miR-127* and *Rian* express lowly in the WT placenta, the color rendering time are longer than others when doing the in situ hybridization. The background color are darker than others, but the signals in dark blue can be seen according to the arrows.

All the images with yellow background in Supplementary Figures were the original images. We have uniformly adjusted the white balance of these images, and hope they look better than before.

Revised figure: Supplementary Fig.2-3

Reviewer #2 (Remarks to the Author):

In this manuscript, the authors focused on the *Gtl2*-*Dio3* region, one of the paternally imprinted genes, and created mice in which polyA sequences were inserted into the *Gtl2* gene to suppress the expression of downstream genes. Gene expression and DNA methylation were abnormal when the PolyA knock-in allele was the maternal allele (MKI) or homozygous (Homo), resulting in abnormal placenta formation and embryonic lethality. The developmental phenotype of the placenta is clear and interesting. Moreover, changes in gene expression and DNA methylation are evident around the knock-in region. However, the RNA sequencing analysis is inadequate, and the problem is that there is no causal link between polyA knock-in, methylation changes, gene expression changes, and placental abnormalities. Specific comments are noted below.

We do appreciate your recognition of our work and professional comments about it. We have revised and improved our manuscript according to your constructive feedback. Our responses to the specific comments are as follows.

Major Comments

1. Line 181: Is there any experimental evidence showing the "clogging blood vessels"? Evidence from histology or alternatives should be provided.

Response:

Thank you for your problem. Our research can not prove the MKI and HOMO fetal blood vessels being clogged after examining our histological sections carefully. This sentence is not rigorous enough. We speculate that the vasculature defect in MKI and HOMO labyrinth maybe result in clogging blood vessels, and then affect the exchange functions of fetal capillaries. This idea may guide us or other researchers to do the related studies about the functions of *Gtl2* polyA knock-in placentas in the future. So we add "maybe" in this sentence in the revised manuscript.

Lines in revised manuscript: 190

Revised text: 'The narrow Fbs in MKI and HOMO maybe result in the accumulation of fetal erythrocytes within them, leading to fetal blood vessels being clogged.'

2. Figure 3: RNA-seq should be analyzed in greater detail. For example, a detailed characterization of each cluster of DEGs should be included. It is strange why the authors did not focus on the commonly changed genes between MKI and HOMO compared to WT. Which kind of GO term was enriched in these common DEGs? It may not be a good idea to analyze up- and down-regulated genes together, since the implications of these DEGs are different. The representative examples of differentially expressed genes/miRNA/lncRNAs should be shown in snapshots or bar charts. What is the sample size for RNA-seq analysis? The authors should analyze at least two biological replicates because the developmental stage should be taken into account. To call DEGs, add a p-value, not just fold changes. This dataset can help the authors explain why the junction zone is expanded in MKI and HOMO from E12.5 to E14.5.

Response:

Following your helpful advice, we have reanalyzed the RNA-seq data detailedly and focused on the enrichment analysis.

Firstly, we have reidentified the DEGs by edgeR (Fig.3a-c). The default parameters of edgeR were used, and DEGs were selected based on log₂ fold changes ≥ 1 and q-values > 0.05 . Because there were no replicates in this study, the biological coefficient of variation (BCV), which is the square-root of dispersions, was set to 0.01 following the suggestion of the edgeR official manual ^[2]. In the volcano map displaying the DEGs, we labeled the representative genes with significant changes in log₂ fold changes. Interestingly, the protein-coding genes in the *Dlk1-Dio3* domain, *Dlk1* and *Rtl1* were the two upregulated DEGs in both MKI and HOMO. *Rtl1* which is crucial for maintaining placental vasculature had the most significant P-value in MKI and HOMO.

Secondly, we overlapped the upregulated DEGs and downregulated DEGs separately, to display the distribution of DEGs (Fig.3d-e). MKI and HOMO had more upregulated DEGs, and PKI had more downregulated DEGs. Maybe upregulated DEGs were likely related to the phenotypes.

Separate enrichment analysis of upregulated and downregulated DEGs can provide a more accurate understanding of the pathways and biological processes. Thirdly, after doing KEGG and GO enrichment with upregulated and downregulated DEGs separately (Supplementary Fig. 4-5), we focused on these terms that were significant in MKI and HOMO, not significant in PKI. For those KEGG terms enriched by upregulated DEGs, 'Cytokine-cytokine receptor interaction', 'ECM-receptor interaction' and 'Hematopoietic cell lineage' were quite related to MKI and HOMO phenotypes and provided important clues about changes in pathway (Fig. 3f). Moreover, we did the enrichment using the upregulated or downregulated DEGs overlapped in MKI and HOMO directly (Fig. 3g-h, Supplementary Fig. 5d and 5h). The GO terms, 'Negative Regulation Of Trophoblast Cell Migration' and 'Regulation Of Morphogenesis Of A Branching Structure' could be enriched (Fig.3g). For KEGG pathways, 'Hematopoietic cell lineage' and 'ECM-receptor interaction' could be enriched, in which all the DEGs were predicted to regulate by the miRNAs in the *Dlk1-Dio3* domain. So, we think MKI and HOMO are highly likely to upregulate those DEGs in KEGG pathways through miRNAs and are the main cause of the phenotypes.

Proteins are the executors of biological functions, so we have directly used differentially expressed mRNAs to do transcriptome analysis and removed the information about miRNAs and lncRNAs in Fig3 to avoid interference by too much information.

We have tried to analyze all the DEGs of PKI, MKI and HOMO together, and then to find the characterization of each cluster in enrichment. But this method cannot provide useful clues about the known phenotypes. We give up using this method to explore the DEGs deeply. The result is shown below.

Lines in revised manuscript: 196-255, 604-612

Revised text: Please see the revised manuscript.

Revised figure: Fig.3, Supplementary Fig.4-6

3. Line 189-190: What is the possible explanation for more DEGs in PKI than MKI (882 vs 294), even though little phenotype is observed in PKI? According to Sup Fig 4A, several GO terms were significantly enriched in comparison to MKI. Does this result suggest the PKI has some gene expression changes independent of imprinting?

Response:

After reidentifying the DEGs and doing enrichment analysis, the PKI (1420) still had the most number of DEGs than MKI (721) and HOMO (1268). Many GO and KEGG terms can be enriched by upregulated or downregulated DEGs (Supplementary Fig. 4a, 4d, Supplementary Fig. 5a, and 5e). Although so many DEGs are in PKI placenta, those DEGs are not enough to function in altering three-layer placental architecture and generating the abnormal vasculature in the labyrinth. While the fewer DEGs in MKI may be the key genes in phenotypes. The gene expression, imprinting and DNA methylation in the *Dlk1-Dio3* domain don't have any change in PKI placenta. We think PKI regulate the DEGs independent of the *Dlk1-Dio3* domain and in the ways we don't know yet. The findings of this study are mainly on the common points in lethal MKI and HOMO. We believe that there must be some changes in PKI placenta development and function that we have not yet noticed. Our research group will investigate PKI placenta in other ways in future and we hope to get something interesting.

4. Line 407-409: "MKI and HOMO embryos with a global *Gtl2* polyA knock-in died after approximately E12.5 accompanied by multiple phenotypes in different organs in parallel, including the placenta, liver, and heart, and so on." No phenotype information of the liver and other tissues is provided in the manuscript. An embryonic *Gtl2* function could be revealed by the tetraploid-chimera-based rescue of placental abnormality.

Response:

In the discussion, we want to express that it seems that the abnormality of the placenta may be one reason for the embryonic lethality, but not the only one. Because our model mice are global *Gtl2* polyA knock-in, our research group is studying various tissues of *Gtl2* polyA knock-in mice, and we have found some abnormal phenotypes in the embryonic body weight, liver and heart. However, those works are still ongoing, and the data have not been published yet. Please forgive us for removing the part about those tissues in this article.

Thank you so much for the valuable advice of rescuing placental abnormality by tetraploid complementation. After finishing accurately figuring out the phenotypes and reasons for the abnormal embryonic tissues, we will definitely consider your advice and discuss it in our research group. To be honest, we have considered the conditional knock-in model mice, but it is difficult for us due to the long construction cycle of model mice and many types of mice used. The tetraploid-chimera-based rescue of placental abnormality may be more suitable for us. It can demonstrate the effect of *Gtl2* polyA knock-in in the embryo in a short period of time. It will be very beneficial for our research in the future.

Lines in revised manuscript: 470-482

Revised text: Please see the revised manuscript.

5. Figure 6: Why are the IG-DMR and *Gtl2*-DMR hypermethylated in MKI and Homo mice? Would this indicate that *Gtl2* 3' ncRNAs play a role in imprint maintenance? In the maternal *Meg3*-DMR deletion mouse, loss-of-imprinting and embryonic lethality were observed, which is similar to the current manuscript (PMID: 31301299). However, *Meg3*-DMR deletion mice did not exhibit altered DNA methylation. It is important to analyze DNA methylation and gene expression at earlier time points (e.g., E6.5 ectoplacental cone and E10.5 early placenta) to gain a better understanding of the mechanism.

Response:

Gtl2 lncRNA can bind to PRC2^[3] components (Ezh2, Eed, and Suz12) directly, and then they inhibit interaction of Ezh2/PRC2 at the IG-DMR locus and subsequent deposition of H3K27me3, which is associated with allelic DNA methylation. The presence of Ezh2/PRC2 in association with *Gtl2* lncRNA prevents Dnmt3s recruitment and subsequent de novo DNA methylation at IG-DMR, and ultimately keeps unmethylation in the maternal allele, and drives expression of the maternal *Gtl2-Rian-Mirg* locus^[4]. The insertion of polyA signals prevented the transcription of *Gtl2* in MKI and HOMO placentas. In the absence of *Gtl2* lncRNA, maybe it is unable to prevent recruitment of Dnmt3s at the IG-DMR locus. Dnmt3s are then recruited to the IG-DMR and deposit de novo DNA methylation, leading to DNA methylation in the maternal IG-DMR.

The methylation of *Gtl2*-DMR in the maternal in the MKI and HOMO placentas may be due to the methylation of IG-DMR in the maternal. IG-DMR acts as the main ICR in the *Dlk1-Dio3* domain, and regulates the methylation status of secondary ICR and the imprinting status in the domain^[5]. Hypomethylated maternal IG-DMR functions as an enhancer to activate

MEGs transcription. When the hypomethylated maternal IG-DMR is methylated through CRISPR/Cas9 based targeted DNA methylation editing tools, the entire IG-DMR is hypermethylated and maternal *Gtl2* is repressed, *Dlk1* is expressed biallelically, and *Gtl2*-DMR gains methylation on the maternal allele^[6]. Similarly, in this study, lacking maternal RNA transcriptions in MKI and HOMO placentas may regulate the methylation status of *Gtl2*-DMR and the imprinting status of *Dlk1* through IG-DMR in the same mechanism.

Besides maternal RNA transcriptions regulate *Dlk1-Dio3* domain through IG-DMR, *Gtl2* lncRNA can control the imprinting of *Dlk1* directly. *Gtl2* lncRNA can partially retain in cis and overlap the maternal *Dlk1*, and prevent the deposit of H3K4me3 and H3K27ac, and then prevent *Dlk1* activation on the maternal allele^[7].

In Wende Zhu's research^[8], loss-of-imprinting in maternal *Meg3*-DMR deletion mouse (*Meg3* ^{$\Delta(1-4)/+$}) placentas and embryonic lethality were observed, which are similar to our MKI and HOMO mice. *Meg3* ^{$\Delta(1-4)/+$} placentas did not exhibit altered DNA methylation, which is quite different from our MKI and HOMO placentas. On the one hand, we use different methods to establish model mice. We operated on different regions of the genome, and maybe the effects on the genome are quite different. We think it is hard to compare the conclusions in these two studies. On the other hand, these two studies have different methods of experimental and statistical analysis of DNA methylation. In their research, after bisulfite treatment, the converted genomic DNAs were amplified by PCR, and then the products were processed into sequencing libraries and sequenced. They did not distinguish the parental alleles and only calculated the overall methylation status which cannot reflect the DNA methylation status on parental alleles. So when the DNA methylation of maternal or paternal allele has a slight change or both alleles have change at the same time, it's hard to recognize the difference in the overall methylation status. While, in our study, after bisulfite treatment, we amplified the converted genomic DNAs, purified the PCR product, and performed TA cloning and sequencing. We distinguished the paternal allele clones from maternal clones by SNPs, and displayed DNA methylation status of each clone after comparing the sequencing sequences and original sequences. In the WT and PKI placentas, IG-DMR is hypermethylated in the paternal allele and hypomethylation in the maternal allele. Our results illustrated that in MKI and HOMO placentas, IG-DMR was hypermethylated in the paternal allele, at the same time, it became hypermethylated in the maternal allele, too. Our method focused on the DNA methylation in paternal allele and maternal allele, separately, and presented them clearly.

We think analyzing DNA methylation and gene expression at earlier time points is important to enrich our study, but is not necessary for the underlying mechanism. Although the results of one placenta of each genotype at E12.5 were shown in the manuscript, the results were not just obtained from one single placenta. Here we present the results of the other two biological replicates to ensure the reproducibility.

Lines in revised manuscript: 407-431

Revised text: Please see the revised manuscript.

Minor Comments

6. Figure 1A-B: Gtl2 locus gene expression changes in Poly-A knock-in mice were not verified in Figure 1. While the authors confirm the silencing of MEGs in Gtl2 locus in MKI and HOMO placentas in Figure 4, conducting the phenotype analysis is logically unclear without confirming gene expression change. The reviewer suggests moving the RT-qPCR results from Figure 4c,f,i to Figure 1.

Response:

Thank you for your suggestion to improve the logical order and readability of the article. On the one hand, we think all the results in Figure 4 should be exhibited as a whole to demonstrate gene expression in *Dlk1-Dio3* domain. On the other hand, following Figure 4, Figure 5 exhibits the changes in spatial expression of some genes in Figure 4 and Figure 6 explains why these gene expression changes. They are coherent up and down. After considering it carefully, we want to keep the original order.

7. Line 169 and 173: Figure 2F appeared before Figure 2E. The figure order should be reorganized.

Response:

Thank you for pointing this out. They have been reorganized in the right order in the revised manuscript and Figure2 panel.

Lines in revised manuscript: 177, 183, 876-885

Revised text: Please see the revised manuscript.

Revised figure: Fig.2e-f

8. There is no consistency in the nomenclature of Gtl2 throughout the manuscript; Gtl2, Gtl2/Meg3, and Meg3 are used randomly. The official gene symbol is Meg3, although Gtl2 has historically been used widely. Therefore, this reviewer believes that using Gtl2/Meg3 rather than Gtl2 would be beneficial to readers.

Response:

In the previous studies, this lncRNA was usually called *Gtl2* when it was studied in mouse, and it was usually called *Meg3* when it was studied in human. The old version manuscript followed this principle. To keep the consistency, we decide to use "*Gtl2*" in mouse, "*Gtl2/Meg3*" in human in the revised manuscript.

Lines in revised manuscript: 442, 443, 444

Revised text: 'Gtl2/Meg3'

9. Lines 433-434: Information about the strain is elusive. It seems likely that the authors generated KI mice with a B6 background first, then crossed them with DBA to produce BDF1 KI mice for allele-specific expression analysis. How about other experiments? Did authors always use mice with DBA backgrounds? This part needs to be clarified.

Response:

Based on the Easi-CRISPR technology, we prepared the sgRNA, ss DNA donor, and Cas9 protein for microinjection. We used pronuclear injection to edit the zygotes of BDF1 strain mice. (The BDF1 mice that have good quality and quantity of zygotes, and can overcome low inbreeding fertility, so BDF1 mice are widely used to establish transgenic mouse model.) We surgically transplanted blastocysts into the uterus of surrogate mice, and then successfully obtained F0 generation mice with positive inserts. One male of them was chosen as the founder of *Gtl2* polyA knock-in mice.

The founder crossed with DBA mice to produce many *Gtl2* polyA knock-in mice with DBA background after several generations. We did all the experiments, except DNA methylation and allele-specific expression analysis using the mice with DBA background.

The founder crossed with B6 mice to produce many *Gtl2* polyA knock-in mice with B6 background after several generations. The mice with DBA background mated with the mice with B6 background to do the DNA methylation and allele-specific expression analysis.

Lines in revised manuscript: 501-511

Revised text: 'Based on the Easi-CRISPR technology, single guide RNA (sgRNA), single strand DNA donor (ssDNA donor), and Cas9 protein were prepared for microinjection. The sequences of sgRNA and ss DNA donor were described in Supplementary Table 2. After pronuclear injection to edit the zygotes of BDF1 (B6 × DBA) strain mice and surgically

transplanting blastocysts into the uterus of surrogate mice, the founder with positive insert was successfully obtained.

The founder mouse was backcrossed into DBA and B6 genetic background, respectively, to gain Gtl2 polyA knock-in mice with DBA background and B6 background. For imprinting analysis and methylation analysis, the Gtl2 polyA knock-in mice in DBA background were used to mate with those in B6 background. All other experiments used Gtl2 polyA knock-in mice in DBA background.'

Reviewer #3 (Remarks to the Author):

In this manuscript, Zhang et al. examined the role of maternal RNA transcription in the Dlk1-Dio3 imprinted domain in placental development. Taking advantage of the feature that the transcripts in the maternal allele in this domain are generated by polycistronic transcription from the Gtl2 promoter, they inserted a polyA cassette at the 3'-end of the Gtl2 promoter and generated a mouse model in which the expression of MEGs (Gtl2, Rian, Mirg and miRNAs) is downregulated, while the Gtl2-DMR sequence is retained and the genome context is minimally altered. They reported that maternal allele polyA knock-in (MKI) and both alleles polyA knock-in (HOMO) embryos died in mid-gestation. In MKI and HOMO placentas, they found that proportion of junctional zones and labyrinth areas were altered and that vascular formation in the labyrinth was abnormal. They also detected de-repression of PEGs (Dlk1 and Rtl1), as well as aberrant DNA methylation of the Gtl2-DMR and IG-DMR on the maternal allele in MKI and HOMO placentas. They did not observe these changes under paternal knock-in (PKI). They concluded that maternal RNA transcription in the Dlk1-Dio3 domain is essential for proper placentation.

The experiments were nicely done and the manuscript was clearly written. The results of this study, which were capable of demonstrating that maternal RNA transcription in the Dlk1-Dio3 domain is required for normal placentation without changing the genome sequence as much as possible, are significant for understanding the regulation of both ontogeny and genomic imprinting. However, due to the altered expression levels of multiple MEGs and PEGs in the domain, the molecular mechanisms of which gene products (proteins, lncRNAs and miRNAs) directly or indirectly regulate placental development have not been elucidated. As the authors discuss, it would be useful to generate mouse models with polyA cassettes inserted downstream of each MEG in future studies. Below I have listed several points, in which the manuscript needs to be improved.

We sincerely appreciate your positive comments. Thank you for the time and effort that have put into reviewing the manuscript, your professional advice have enabled us to improve our work.

1) Lines 98-101, Fig. 1A

Please indicate the chromosomal position (coordinate) where the polyA cassette was inserted. Sequence information of the donor DNA fragment used in CRISPR genome editing should also be provided.

Response:

Thank you so much for your advice. The chromosomal position of the polyA signals inserted: chr12: between 109,541,067 and 109,541,068.

The single strand DNA donor (ssDNA donor, 341bp) contains three parts, including left homo-arm (98bp), insert segment (3x polyA, 147bp) and right homo-arm (96bp).

The sequence of the ssDNA donor:

```
ATATAAACCCACCCAGCCAGCCCTAGCACAGAAGACGAAGAGCTGGAATAGAGCTCGCCTC
GGCTCTGCTGGCCTTGGCTGCAGCTCTCCAGAAAAATAAAAAGATCTTTATTTTCATTAGATCTG
TGTGTTGGTTTTTGTGTGAATAAAAGATCTTTATTTTCATTAGATCTGTGTGTTGGTTTTTGTGTG
AATAAAAGATCTTTATTTTCATTAGATCTGTGTGTTGGTTTTTGTGTGCCCGGGCGCCACAGA
AGAATCTCTTACCTGGTGAGTGGTTAGCCATCCTTGCCTGAAAGGATGTGCAAAAATGAAGAC
GACATCACTATCTGG
```

Lines in revised manuscript: 499-500

Revised text: 'The chromosomal position of insertion: mouse (GRCm38/mm10) chr12: after 109,541,067.'

Revised table: Supplementary Table 3 The sequences of sgRNA and ssDNA donor.

2) Lines 133-136 and lines 407-409

The body weight, liver, and heart phenotypes of *Gtl2* polyA knock-in embryos are mentioned, but no relevant data are presented. They should be presented.

Response:

In the discussion, we want to express that it seems that the abnormality of the placenta may be one reason for the embryonic lethality, but not the only one. Because our model mice are global *Gtl2* polyA knock-in, our research group is studying various tissues of *Gtl2* polyA knock-in mice, and we have found some abnormal phenotypes in the embryonic body weight, liver and heart. However, those works are still ongoing, and the data have not been published yet. Please forgive us for removing the part about those tissues in this article.

Lines in revised manuscript: 137-139

Revised text: 'MKI and HOMO placentas had no remarkable differences during the period before death, which suggests that MKI and HOMO placenta may be not responsible for embryonic lethality.'

Lines in revised manuscript: 470-482

Revised text: Please see the revised manuscript.

3) Lines 165-173

Figure 2 panels presented out of order (Fig. 2E is explained after 2F).

Response:

Thank you for pointing this out. They have been reorganized in the right order in the revised manuscript and Figure2 panel.

Lines in revised manuscript: 177, 183, 876-885

Revised text: Please see the revised manuscript.

Revised figure: Fig.2e-f

4) Fig. 6A

Allelic-specific expression of the *Dlk1* gene should be quantified (e.g. by pyrosequencing) for multiple samples (biological replicates).

Response:

Thank you again. Quantifying allelic-specific expression of *Dlk1* in different biological replicates is a good idea to show the change in the expression of *Dlk1*. After consideration, we have decided to solve this problem with strand-specific RT-PCR^[9,10] to detect expression level of *Dlk1* in both stands. In the WT and PKI placentas, *Dlk1* transcribed from only one strand, but in MKI and HOMO placentas, *Dlk1* transcribed from both strands. Semi-quantitative PCR showed *Dlk1* had a similar expression level in both stands.

The cDNA synthesis reaction was primed with oligonucleotides complementary to *Gapdh* (*Gapdh*-R: 5'- GCATGGACTGTGGTCATGAG - 3'), *Dlk1* antisense (*Dlk1*-F: 5' - ACGGGAAATTCTGCGAAATA - 3'), and *Dlk1* (*Dlk1*-R: 5' - CTTCCAGAGAACCCAGGTG - 3'). PCR was performed with primers to detect transcripts for *Gapdh* (*Gapdh*-F: 5' - CCATCACCATCTTCCAGGAG - 3' and *Gapdh*-R), *Dlk1* antisense and *Dlk1* (*Dlk1*-F and *Dlk1*-R). Biological replicates: n=3 for each genotype.

5) Fig. 6B

In the *Dlk1*-*Dio3* domain map, the insertion site of the polyA signal and the regions analyzed by bisulfite sequencing should be illustrated to help better understanding.

Response:

We fully agree with your advice, and we have added the insertion site of the polyA signal in Figure 6B to improve the readability of the results.

Revised figure: Fig.6b

6) Figs. 6C-E

Was DNA methylation analysis of each genotype by bisulfite sequencing conducted using a DNA sample extracted from only one placenta? To ensure reproducibility of data, it is important to employ biological replicates and the results for multiple samples should be presented.

Response:

Our DNA methylation results are not obtained from one placenta. We have done the experiments of three placentas each genotype. The manuscript only displays the result of biological replicate 1. Here we present the results of the other two biological replicates to ensure the reproducibility.

7) Figs. 6C-E

Gtl2 polyA knock-in caused loss-of-imprinted DNA methylation (gain of methylation on the maternal KI allele) in Gtl2-DMR and IG-DMR, but not in Dlk1-DMR. Discussion on the molecular mechanism of this phenotype should be included.

Response:

Thank you for your valuable advice. We have discussed about the possible molecular mechanism of the changes in IG-DMR and Gtl2-DMR and unchange in Dlk1-DMR in the third paragraph in the discussion section of the revised manuscript. We hope they can help the reviewers and the readers understand our study better.

Lines in revised manuscript: 407-431

Revised text: Please see the revised manuscript.

8) Line 410

The "brian" should be corrected to "brain".

Response:

Thank you for pointing this spelling mistake out, and it has been corrected in the revised manuscript.

Lines in revised manuscript: 477
Revised text: 'placenta-brain axis'

9) Supplementary Table 1

Please check that the primer sequences for DNA methylation analysis are correct. Some primers contain both "C" and "G" in the sequence, which is unusual for bisulfite sequencing primers.

Response:

Thank you to help us to find this mistake. Due to our negligence, we copied the genome sequences of the methylation primers of *Gtl2*-DMR. We have corrected the wrong sequences in the revised manuscript, and examined all primer sequences carefully.

Revised table: Supplementary Table 4 Primer sequences.

References

1. Elmore, S.A., et al., *Histology Atlas of the Developing Mouse Placenta*. Toxicol Pathol, 2022. **50**(1): p. 60-117.
2. Zhang, M., et al., *A peptide encoded by circular form of LINC-PINT suppresses oncogenic transcriptional elongation in glioblastoma*. Nature Communications, 2018. **9**(1).
3. Zhao, J., et al., *Genome-wide Identification of Polycomb-Associated RNAs by RIP-seq*. Molecular Cell, 2010. **40**(6): p. 939-953.
4. Das, P.P., et al., *PRC2 Is Required to Maintain Expression of the Maternal Gtl2-Rian-Mirg Locus by Preventing De Novo DNA Methylation in Mouse Embryonic Stem Cells*. Cell Rep, 2015. **12**(9): p. 1456-70.
5. Saito, T., et al., *A tandem repeat array in IG-DMR is essential for imprinting of paternal allele at the Dlk1-Dio3 domain during embryonic development*. Human Molecular Genetics, 2018. **27**(18): p. 3283-3292.
6. Kojima, S., et al., *Epigenome editing reveals core DNA methylation for imprinting control in the Dlk1-Dio3 imprinted domain*. Nucleic Acids Res, 2022. **50**(9): p. 5080-5094.
7. Sanli, I., et al., *Meg3 Non-coding RNA Expression Controls Imprinting by Preventing Transcriptional Upregulation in cis*. Cell Rep, 2018. **23**(2): p. 337-348.
8. Zhu, W., et al., *Meg3-DMR, not the Meg3 gene, regulates imprinting of the Dlk1-Dio3 locus*. Dev Biol, 2019. **455**(1): p. 10-18.
9. Riordan, J.D., et al., *Identification of rtl1, a retrotransposon-derived imprinted gene, as a novel driver of hepatocarcinogenesis*. PLoS Genet, 2013. **9**(4): p. e1003441.
10. Sekita, Y., et al., *Role of retrotransposon-derived imprinted gene, Rtl1, in the fetomaternal interface of mouse placenta*. Nat Genet, 2008. **40**(2): p. 243-8.

Reviewers' comments:

Reviewer #1 (Remarks to the Author):

The authors have satisfactorily addressed the previous concerns.

Reviewer #2 (Remarks to the Author):

According to this reviewer, the authors have attempted to respond to most comments raised during the initial review. However, despite improvements in GO analysis, the RNAseq analysis shown in the revised manuscript still has a potential problem.

Major comments

1. Even with the authors' explanation in the rebuttal letter, this reviewer could not understand how to calculate the p-value with only one sample in each genotype group. Have the authors included technical replicates in the analysis? How does "set" the BCV work? Additionally, since two of the significantly altered genes, *Uty* and *Kdm5d*, are Y-linked genes, this reviewer suspects only WT is female, while MKI, PKI, and Homo are males.

Minor comments

1. Line 244-245: Since miRNAs regulate translation, down-regulation detected in RNAseq, which represents a transcription change, cannot be explained by miRNA down-regulation.
2. Line 264: Delete "mRNA."
3. Lines 417-420: Cite corresponding references (PMID: 35544282?).
4. Lines 903-904: Describe what the red dashed line indicates in Figure 3f.
5. Figure 4a, d, and g: These figures should include genomic locus information at the bottom or top so readers can see which loci are affected.
6. Figure 4h: It would be good if the authors included the miRNA expression in other loci than *Gtl2* locus as a control to prove either that the entire miRNA expression machinery was affected in MKI and HOMO or that only miRNA in *Gtl2* locus was affected.
7. The authors should discuss the inconsistency with Wende Zhu's research in the discussion. Despite disagreeing with the author's claim that the methylation analysis methodology causes the different results, the reviewer can agree that it is due to differences between transgenic mice, polyA-Ki and deletion. In the discussion, it would be fair to describe.
8. The strand-specific RT-PCR and bisulfite sequences of biological replicates, shown only in the rebuttal letters, are quite informative for readers. Those figures should be included in the supplementary materials.

Reviewer #3 (Remarks to the Author):

The authors have responded to my comments in their revised manuscript. I have no further concerns.

Point-by-Point Response to Reviewers (2 nd revision)

Reviewers' comments:

*Responses to the reviewers' comments are highlighted in blue.

Reviewer #2 (Remarks to the Author):

According to this reviewer, the authors have attempted to respond to most comments raised during the initial review. However, despite improvements in GO analysis, the RNAseq analysis shown in the revised manuscript still has a potential problem.

Thank you very much for taking your time to review this manuscript. Thank the reviewer for the valuable comments contributing to the improvement of the manuscript.

Major comments

1. Even with the authors' explanation in the rebuttal letter, this reviewer could not understand how to calculate the p-value with only one sample in each genotype group. Have the authors included technical replicates in the analysis? How does "set" the BCV work? Additionally, since two of the significantly altered genes, Uty and Kdm5d, are Y-linked genes, this reviewer suspects only WT is female, while MKI, PKI, and Homo are males.

Response:

In alignment with the methodologies employed in prior studies [1-3], the edgeR package [4] was utilized for the computation of p-values pertaining to differential gene expression in the absence of replicate samples. Due to the absence of replicate samples in this study, following the guidelines outlined in the edgeR official manual, we set the Biological Coefficient of Variation (BCV), utilized for conducting exact test [5]. This allowed us to draw statistical significance conclusions (i.e., p-values) regarding differential expression.

In exact tests, the square of the Biological Coefficient of Variation (BCV) serves as the input for dispersion [4]. Within edgeR, the variability in the data is initially addressed by estimating the dispersion, which reflects the degree of gene expression heterogeneity within samples. Subsequently, based on these estimated dispersions, exact test [5] are employed to calculate p-values for differential expression. In scenarios with replicate samples, calculation can be performed by the differences in the true abundance of genes between replicated RNA samples. Conversely, in the absence of replicate samples, one can directly utilize the BCV value recommended by edgeR.

All in all, in the absence of replicates, the principle of exact test entails the direct comparison of two samples between the control and treatment groups. This method employs a simulation-based approach to calculate the probability of observing differences in expression levels within these two groups. The simulation accounts for the given biological variations, assessing whether the observed disparities in gene expression are sufficiently extreme to permit inference regarding the significance of differential expression for a specific gene.

For Y-linked genes, we didn't consider the effect of sex when preparing the experimental

samples for the RNA-seq sequencing, and didn't know the sex of those four samples at that time. However, based on the genomic PCR result of *Sry* (sex determining region Y), and gene expression of Y-linked genes (*Uty* and *Kdm5d*), WT sample is female, PKI, MKI and HOMO samples are males. Here, we have identified the sex of those four samples using genomic DNA isolated from the tails of the embryos.

Following your speculation, we also have done semi-qRT-PCR to verify the expression of *Uty* and *Kdm5d*.

Next, we have re-done RNA-seq analysis after deleting genes in Y chromosome to eliminate the influence of sex. (We have annotated the chromosome numbers of all DEGs in PKI, MKI, and HOMO. They are listed in Supplementary Data 1 Fig.3a, Fig.3b and Fig.3c.) There are only two DEGs (*Uty* and *Kdm5d*) in Y chromosome in PKI, MKI and HOMO, also they are not included in MKI and HOMO overlapped genes that are related to the phenotypes for GO analysis. And, we don't do any gene editing in Y chromosome in the model mice. Those indicate that Y-linked DEGs are not related to the phenotypes in *Gtl2* polyA knock-in placentas.

In addition, it is difficult for us to prepare enough samples of each genotype for RNA-seq in a short time. For MKI and HOMO placentas, firstly, we need to use WT females to mate with PKI males to get many age-appropriate female PKI mice with good physiological conditions. Then, these PKI females need to mate with PKI or WT males and try to be pregnant at the same time as possible. A pregnant female DBA/2J mouse gives birth to 4-5 offspring per litter, not like other strains that can have many offspring per litter. The placentas and their embryos in different litters should have the same developmental stage and the same size. Also, winter is not a good season for mice to breed. We need at least many months to prepare those mice. So, our experimental samples are very valuable, and it is hard to get several of each genotype at the same time for re-doing RNA-seq.

On the other hand, we have further verified fourteen candidate genes by qRT-PCR using having collected independent biological replicates (Supplementary Figure 4). We have verified four DEGs (two upregulated and two downregulated) with significant changes in log₂ fold-change, high expression levels and not in the sex chromosomes of each genotype. Also, in the volcano in Figure 3a-c, we have labeled the DEGs that are significant and verified as representative examples instead of just significant DEGs. And we think MKI and HOMO overlapped upregulated DEGs are related to the phenotypes. So two MKI and HOMO overlapped upregulated DEGs are verified. All the results of qRT-PCR of candidate DEGs have the same trends of change with RNA-seq results. Besides, the results in Figure 4 (qRT-PCR of genes (mRNAs, lncRNAs, and miRNAs) in chromosome 12) can also help to verify the RNA-seq data. We hope they can help to explain the concern about RNA-seq.

Lines in revised manuscript: 187-188, 190-193, 214

Revised text:

'There were 1418 DEGs (up:377, down:1041) in PKI, 719 DEGs (up:398, down:321) in MKI and 1266 DEGs (up:875, down:391) in HOMO (Fig.3a-c).'

'Some DEGs with significant changes and high expression levels were verified by qRT-PCR in Supplementary Fig.4a-c.'

'Two overlapped upregulated DEGs were verified by qRT-PCR in Supplementary Fig.4d.'

Revised figures: Supplementary Figure 4, Figure 3a-e

Minor comments

1. Line 244-245: Since miRNAs regulate translation, down-regulation detected in RNAseq, which represents a transcription change, cannot be explained by miRNA down-regulation.

Response:

We sincerely appreciate you for pointing out our problem that we were not aware of. Changes in gene transcription level cannot be explained by changes in miRNA expression. We have deleted the related words to ensure the accuracy of the article.

Lines in revised manuscript: 220-224, 578

Revised text: Please see the revised manuscript.

2. Line 264: Delete "mRNA."

Response:

This word (mRNA) here functions as a subheading, and we mainly want to tell the readers that this paragraph is about the expression of mRNAs in *Dlk1-Dio3* domain. "lncRNA" in line 244 and "miRNA" in line 250 have the same meaning. We have deleted them together in the revised manuscript.

Lines in revised manuscript: 232, 244, 250

Revised text: Please see the revised manuscript.

3. Lines 417-420: Cite corresponding references (PMID: 35544282?).

Response:

Thank you for reminding. We have inserted the corresponding reference.

Lines in revised manuscript: 395-399, 754-756

Revised text: 'Kojima, S. et al. Epigenome editing reveals core DNA methylation for imprinting control in the Dlk1-Dio3 imprinted domain. Nucleic Acids Res 50, 5080-5094, doi:10.1093/nar/gkac344 (2022).'

4. Lines 903-904: Describe what the red dashed line indicates in Figure 3f.

Response:

The red dashed line represents the P.value is 0.05. $\log_{10}0.05$ is approximately equal to 1.3. The terms above that line are significantly enriched and the terms below are not significantly enriched. We have described the red dashed line in the revised manuscript.

Lines in revised manuscript: 869-870

Revised text: 'The red dashed line represents the p is 0.05.'

5. Figure 4a, d, and g: These figures should include genomic locus information at the bottom or top so readers can see which loci are affected.

Response:

We thank the reviewer for helping us improve the details of the figure. We have added the detailed genomic locus at the bottom.

Revised figures: Figure 4a, d, and g

6. Figure 4h: It would be good if the authors included the miRNA expression in other loci than Gtl2 locus as a control to prove either that the entire miRNA expression machinery was affected in MKI and HOMO or that only miRNA in Gtl2 locus was affected.

Response:

Following your helpful advice, we have added the expression in four genotypes of *miR-345* and *miR-1247* located separately before and after *Dlk1-Dio3* domain in Figure 4h. They are the miRNAs closest to the domain and can help to illustrate that only the miRNAs in the domain were affected in MKI and HOMO.

Lines in revised manuscript: 257-259

Revised text: 'While miR-345 and miR-1247 located separately before and after Dlk1-Dio3 domain did not exhibit that expression trend.'

Revised figures: Figure 4h

7. The authors should discuss the inconsistency with Wende Zhu's research in the discussion. Despite disagreeing with the author's claim that the methylation analysis methodology causes the different results, the reviewer can agree that it is due to differences between transgenic mice, polyA-Ki and deletion. In the discussion, it would be fair to describe.

Response:

We totally agree with your advice, and we have added some discussion about the inconsistency with Wende Zhu's research. We believe they will help a lot for further studies about the *Dlk1-Dio3* domain.

Lines in revised manuscript: 377-384

Revised text: 'Maternal Gtl2/Meg3-DMR deletion (Meg3Δ(1-4)/+) mice in Wende Zhu's research exhibited dysregulating gene expression, loss-of-imprinting in the placentas and embryonic lethality, which are similar to our MKI and HOMO mice. Meg3Δ(1-4)/+ placentas did not exhibit altered DNA methylation, which is quite different from our MKI and HOMO placentas.9 Different methods to establish model mice, operating on different genomic regions, and acquisition and deletion of sequences in those two model mice all likely lead to quite different effects on the genome and epigenetics (DNA methylation).'

8. The strand-specific RT-PCR and bisulfite sequences of biological replicates, shown only in the rebuttal letters, are quite informative for readers. Those figures should be included in the supplementary materials.

Response:

Thank you so much for your advice. Our old manuscript only told the readers about the results and conclusions. Following your advice, we have provided the results about strand-specific RT-PCR and bisulfite sequences of biological replicates in the supplementary materials.

Lines in revised manuscript: 305-307 (strand-specific RT-PCR), 349-350 (bisulfite sequences of biological replicates)

Revised text: 'Strand-specific RT-PCR of Dlk1 verified that result and Dlk1 had a similar expression level in both stands in MKI and HOMO (Supplementary Fig.8a).'

'The results of IG-DMR, Gtl2-DMR and Dlk1-DMR in other two biological replicates were provided in Supplementary Fig.8b-d.'

Revised figures: Supplementary Figure.8

References

- [1] Zhang M,Zhao K,Xu X, et al. A peptide encoded by circular form of LINC-PINT suppresses oncogenic transcriptional elongation in glioblastoma. *Nat Commun.* 2018;9 (1):4475. doi:10.1038/s41467-018-06862-2
- [2] Li Y,Tu M,Feng Y, et al. Common metabolic networks contribute to carbon sink strength of sorghum internodes: implications for bioenergy improvement. *Biotechnol Biofuels.* 2019;12:274. doi:10.1186/s13068-019-1612-7
- [3] Huang C,Liang Y,Dong Y, et al. Novel prognostic matrisome-related gene signature of head and neck squamous cell carcinoma. *Front Cell Dev Biol.* 2022;10:884590. doi:10.3389/fcell.2022.884590
- [4] McCarthy, DJ, Chen, Y, Smyth, GK (2012). Differential expression analysis of multifactor RNA-Seq experiments with respect to biological variation. *Nucleic Acids Research* 40, 4288-4297. doi:10.1093/nar/gks042
- [5] Robinson MD and Smyth GK (2008). Small-sample estimation of negative binomial dispersion, with applications to SAGE data. *Biostatistics*, 9, 321-332.

Reviewers' comments:

Reviewer #2 (Remarks to the Author):

It is appreciated that the authors addressed the concerns raised by this reviewer in the revised manuscript. While the issue of not including biological replicates in the RNA-Seq analysis remains, the authors have attempted to increase the reliability of the findings in this revised manuscript by including RT-qPCR. In mice, biological replication is generally mandatory, especially in placentas where growth is highly variable between individuals, as illustrated in Figure 1d-f. Using simulations to estimate variation is not common today in this field, but certain trends can be observed.

There are two minor, but crucial issues that should be addressed in order to further improve the manuscript:

1. In a rebuttal letter to the previous version (but not in this manuscript), the authors mentioned: "the biological coefficient of variation (BCV), which is the square-root of dispersions, was set to 0.01 following the suggestion of the edgeR official manual". However, as far as this reviewer learned, the EdgeR manual states, "Typical values for the common BCV (square-root-dispersion) for datasets arising from well-controlled experiments are 0.4 for human data, 0.1 for data on genetically identical model organisms or 0.01 for technical replicates". Thus, this reviewer believes 0.1 should be used in the analysis in this manuscript.
2. Additionally, there is a lack of sufficient description regarding EdgeR analysis of RNA-seq in this manuscript. Since EdgeR is a simulation-based analysis, it is imperative to provide detailed information about the analysis to enhance transparency.

Point-by-Point Response to Reviewers (2 nd revision)

Reviewers' comments:

*Responses to the reviewers' comments are highlighted in blue.

Reviewer #2 (Remarks to the Author):

It is appreciated that the authors addressed the concerns raised by this reviewer in the revised manuscript. While the issue of not including biological replicates in the RNA-Seq analysis remains, the authors have attempted to increase the reliability of the findings in this revised manuscript by including RT-qPCR. In mice, biological replication is generally mandatory, especially in placentas where growth is highly variable between individuals, as illustrated in Figure 1d-f. Using simulations to estimate variation is not common today in this field, but certain trends can be observed.

Thank you very much for taking time and efforts to review our manuscript. We sincerely appreciate your comments and professional advice. They help to improve academic rigor of our article and guide our revision.

There are two minor, but crucial issues that should be addressed in order to further improve the manuscript:

1. In a rebuttal letter to the previous version (but not in this manuscript), the authors mentioned: "the biological coefficient of variation (BCV), which is the square-root of dispersions, was set to 0.01 following the suggestion of the edgeR official manual". However, as far as this reviewer learned, the EdgeR manual states, "Typical values for the common BCV (square-root-dispersion) for datasets arising from well-controlled experiments are 0.4 for human data, 0.1 for data on genetically identical model organisms or 0.01 for technical replicates". Thus, this reviewer believes 0.1 should be used in the analysis in this manuscript.

Response:

Following your valuable advice, we have studied the edgeR manual carefully, and then set the BCV as 0.1 for reanalyzing the DEGs to make sure the accuracy of the data. DEGs were selected based on \log_2 fold-change > 1 and p -value < 0.01 . We have changed BCV (as 0.1) and kept other default parameters unchanged. The DEG number of each genotype placenta has a little smaller than that in the previous version, but most DEGs overlap with the previous ones. Then, we have updated all the enrichment analysis results using new DEGs. Although the results are not totally the same as the previous ones, most enrichment terms are the same, especially the terms related to the abnormal phenotypes. The new DEGs don't change the conclusion.

*Revised figure: figure3, supplementary figure5, supplementary figure6, supplementary figure7
Lines in revised manuscript: 187-189, 207-210, 213, 216-226*

Revised text:

'There were 1256 DEGs (up:312, down:944) in PKI, 572 DEGs (up:314, down:258) in MKI and 1081 DEGs (up:749, down:332) in HOMO'

'The overlapped upregulated DEGs were more than the overlapped downregulated DEGs (91 vs 40)'

'Regulation Of Mitotic Spindle Checkpoint' functions in the differentiation of placental trophoblast stem cells to trophoblast giant cells 43. Due to placental vasculature has a branching structure and the endothelial cells come from extra-embryonic mesodermal cells-derived allantois 1, 'Regulation Of Morphogenesis Of A Branching Structure', 'Mesodermal Cell Fate Commitment' are likely related to the MKI and HOMO phenotypes. '

2. Additionally, there is a lack of sufficient description regarding EdgeR analysis of RNA-seq in this manuscript. Since EdgeR is a simulation-based analysis, it is imperative to provide detailed information about the analysis to enhance transparency.

Response:

We thank the reviewer so much for pointing out this omission. It is important to describe the detailed methods accurately and clearly, especially the critical parts. Based on your helpful advice, we have added more details about EdgeR analysis of RNA-seq in the "Materials and methods" section. We emphasized there was no biological replication in this study, and the setting of BCV as 0.1 to enhance transparency. There was no specific setting in the remaining steps, and all parameters are default for edgeR.

Lines in revised manuscript: 579-583

Revised text:

'The default parameters of edgeR 71 were used, and DEGs were selected based on log2 fold-change > 1 and p-value < 0.01. Since there was no biological replication in this study, the biological coefficient of variation (BCV), which is the square-root of dispersion, was set to 0.1 according to the suggestion of the edgeR official manual. Functional enrichment analysis was conducted using Enrichr 72. For Biological Process categories, we used Go terms, and other parameters were left at their default settings.'